# Age and Petrogenesis of the Dongjin Rare Metal Mineralized Intrusion in the Northern Margin of the North China Craton

Chenyu Liu [1,2,3], Gongzheng Chen [1,2,3,*], Jinfang Wang [1,2,3], Yi Cheng [4], Kangshuo Li [1], Zeqian Lu [1] and Yutong Song [1]

1  College of Earth Sciences, Hebei GEO University, Shijiazhuang 050031, China; liuchenyu202308@163.com (C.L.); wjfb1983@163.com (J.W.); 17703211393@163.com (K.L.); 13361002809@163.com (Z.L.); songyutong20010211@163.com (Y.S.)
2  Hebei Province Collaborative Innovation Center for Strategic Critical Mineral Research, Hebei GEO University, Shijiazhuang 050031, China
3  Innovation Base for Mineralization Theory and Prospecting Technology of Ophiolite Belt, Hebei GEO University, Shijiazhuang 050031, China
4  The Seventh Geological Brigade, Hebei Bureau of Geology and Mineral Exploration and Development, Langfang 065000, China; 17731404803@163.com
*  Correspondence: gzchen@hgu.edu.cn

**Abstract:** Highly fractionated granites are widespread in the middle part of the northern margin of the North China Craton (MNNCC), and several are accompanied by rare metal mineralization. The Dongjin rare metal mineralized intrusion, which is representative of this region, is composed of fine-grained alkali-feldspar granite (FAG) and kali-feldspar granite (KG). The FAG and KG evolve continuously, exemplifying the relationship between magmatic evolution and rare metal mineralization. In this contribution, we present integrated columbite U-Pb geochronology, mineralogy, and whole-rock geochemistry analyses of the Dongjin intrusion to determine the timing of the mineralization, petrogenesis, and geodynamic setting, from which the following results are obtained: (1) LA-ICP-MS U-Pb dating for columbite of the FAG and KG yielded the lower intercept ages between 248.9 ± 1.9 Ma and 250.1 ± 1.1 Ma on the Tera–Wasserburg concordia diagram; (2) Geochemically, the Dongjin intrusion is characterized by an enrichment in Si, Al, Rb, Th, U, Nb, and Zr and a strong depletion in Ba, Sr, P, and Ti, with extremely negative Eu anomalies, high LREE and HREE values, and a noticeable tetrad effect of rare earth elements; as a result, it belongs to high-K calc-alkaline rocks; (3) The Dongjin intrusion belongs to a highly differentiated I-type or A-type granite; (4) The fractional crystallization of plagioclase, K-feldspar, and biotite occurred during magmatic evolution; (5) The Dongjin intrusion was formed in a post-collisional extensional environment. In conclusion, the FAG and KG have a homologous evolution, and the FAG has a higher degree of fractional crystallization. The enrichment and mineralization of Nb-Ta are related to the highly fractionated crystallization of granitic magma and fluid–melt interactions in the final stages of magmatic evolution, and there is a rare metal mineralization related to highly fractionated granite in the MNNCC in the Early Triassic, which deserves full attention in future research and prospecting.

**Keywords:** columbite U-Pb dating; rare metal mineralization; highly fractionated I-type granite; northern margin of the North China Craton

## 1. Introduction

Growth in the electronics, aerospace, and medical industries has led to increased demand for rare metals such as niobium (Nb), tantalum (Ta), beryllium (Be), zirconium (Zr), and lithium (Li), which are also recognized as "critical metals" [1]. Nb and Ta are essential rare-metal materials in the high-tech electronics industry; therefore, they are widely applied in aerospace and electrometallurgy [2].

Worldwide, Nb-Ta deposits are located mainly in Brazil, Australia, Canada, and China. Nb-Ta mineralization occurs in primary or secondary deposits. Primary deposits are dominantly related to igneous rocks, where the mineralization is either magmatic or hydrothermal in origin and can be divided according to igneous association: (1) Carbonatites: these rocks contain most of the world's niobium resources and are often symbiotic with rare earth elements (REE) (e.g., Bayan Obo in China and Nolans Bore in Australia [3,4]); (2) Peralkaline granitic and silica-undersaturated rocks: the mineralization characteristics in these rocks are related to REE-Y-Nb-Zr concentration, and, in some cases, Ta mineralization also occurs (e.g., the Ghurayyah Ta deposit in Saudi Arabia and the Motzfeld Ta deposit in Greenland [3]); (3) Metaluminous and peraluminous granitic rocks: these rocks are host to the world's major Ta deposits (e.g., the Pitinga Ta-Li deposit in Brazil [3] and the Yichun Ta-Nb deposit in China [5]). Granite-hosted Nb-Ta deposits are closely associated with Sn, and pegmatite-hosted Ta deposits are often mined for Li or Cs [5–7]. Most secondary deposits involve the re-enrichment of Nb-Ta resources after secondary physical and chemical processes in primary deposits, and the most representative ones are carbonate weathering crust-type niobium deposits [3]. In a peraluminous granitic magma system, the host rocks have usually undergone high degrees of fractional crystallization.

Nb-Ta deposits in China are mainly distributed in South China and Xinjiang [6]. With the deepening of prospecting exploration work, several Nb-Ta deposits have been discovered in the southern Great Xing'an Range and the northern margin of the North China Craton (NCC) on both sides of the Solonker suture zone in recent years (e.g., the Zhaojinggou Ta-Nb deposit [6,7], Jiabusi Nb-Ta deposit [8,9], Saima Nb deposit [10,11], Huashi Rb-Ta-Nb deposit [12], and Daxiyingzi Rb-Be-Nb deposit [13,14]), showing substantial ore-forming potential. These rare metal deposits are closely related to highly fractionated granites and possess strong metallogenic specialization [6–14]. Additionally, the MNNCC harbors highly fractionated granites with exceptional metallogenic potential [6,7]. However, significant progress has yet to be achieved in Nb-Ta deposit prospecting for a long time. At present, several rare metal deposits such as Hua'shi and Han'erzhuang have been found [12,15,16].

Many studies have been carried out on the geochronology of rare metal deposits in the northern margin of the NCC. The results show that these deposits were mainly formed from the Late Triassic to the Early Cretaceous. For example, Zhang and Jiang [17] obtained a columbite U-Pb age of $130 \pm 2.0$ Ma for albite granite in the Zhaojinggou Ta-Nb deposit, Inner Mongolia. Zhang [9] carried out cassiterite U-Pb dating for the Jiabusi Nb-Ta deposit and obtained an isochron age of $149 \pm 2.0$ Ma for highly fractionated granite. The typical magmatic zircon U-Pb age of $229.5 \pm 2.2$ Ma for the aegirine nepheline syenite of the Liaoning Saima Nb deposit suggests that the deposit was formed in the Late Triassic [10]. Ju et al. [13] reported that biotite $^{40}$Ar-$^{39}$Ar ages of biotite granite and albitite granite in the Daxiyingzi deposit were $223.37 \pm 2.39$ Ma and $223.37 \pm 2.45$ Ma, respectively. Some scholars considered that the Nb-Ta enrichment mechanism is generally considered to be the result of the highly fractional crystallization of granitic magmas [18–21], whereas others argue that fluid–melt interaction plays a vital role in Nb-Ta mineralization [22,23]. The genesis of these Nb-Ta deposits in the northern margin of the NCC is closely related to the magma differentiation and fluid–melt interaction. For example, Li et al. [7] reported that the rare earth elements in the amazonitized and albitized granite of the Zhaojinggou Ta-Nb deposit exhibit a well-visible tetrad effect, as well as geochemical characteristics with Nb/Ta ratios less than 5, indicating that these ore-forming granites have experienced a high degree of crystallization and fluid–melt interaction. Zhang et al. [9] demonstrated that the rare earth elements in the Li-mica and topazlepidolite granites of the Jiabusi Nb-Ta deposit exhibit a remarkable M-type lanthanide tetrad effect, with low Nb/Ta (<1.2) and Zr/Hf (<5.0) ratios. The Dongjin intrusion is located in the MNNCC, with disseminated Nb-Ta mineralization occurring on the top. Moreover, the FAG and KG in the Dongjin intrusion are zoned, with the FAG in the upper part and the KG in the lower part, which offers a superior example with which to study the relationship between crystallization

and mineralization. To date, there remains a dearth of geochronological research on the Dongjin intrusion, hampering comprehension of petrogenesis and mineralization. The zircons within the Dongjin intrusion exhibit high U content with significant metamictization. We attempted zircon U-Pb dating but failed to obtain reliable ages. As a common accessory mineral in highly fractionated granites, columbite shows limited effects of late hydrothermal alteration while possessing high U and a low abundance of Pb [24–26]. This leads to a highly accurate U-Pb dating of columbite, which has produced convincing results across numerous studies [24–29]. In addition, the relationship between magmatic evolution and the mineralization of the Dongjin intrusion remains unclear, and further study is urgently needed.

Here, we provide systematic columbite U-Pb ages of the FAG and KG. We also carry out mineralogy and whole-rock geochemistry analyses of the Dongjin rare metal mineralized intrusion based on detailed geological investigations. These works allow us to (1) determine the crystallization and metallogenic ages of the Dongjin intrusion; (2) explore the petrogenesis and the genetic relationship between the FAG and KG; (3) identify the relationship between magmatic evolution and Nb-Ta mineralization; (4) provide regional implications for future rare-metal prospecting in the northern margin of the NCC.

## 2. Regional Geology

The NCC is the oldest geological unit in China, which has experienced a long geological evolution with rich mineral resources [30–33]. The NCC is bounded by three young orogenic belts: in the north, the Paleozoic–Early Mesozoic Central Asian Orogenic Belt; in the south, the Early Mesozoic Dabie–Sulu Orogenic Belt; and, in the east, the Mesozoic–Cenozoic Circum-Pacific terrane. The northern margin of the NCC is divided into eastern, central, and western segments by the Tan–Lu and Daxing'anling–Taihangshan faults [34,35].

The exposed strata in the northern margin of the NCC include Archean, Paleoproterozoic, Mesoproterozoic, Neoproterozoic, Lower Paleozoic, Upper Paleozoic, Mesozoic, and Paleogene–Neogene. The crystalline basement in the central western segments is composed of a suite of amphibolite–granulite facies comprising the metamorphic rocks of the Archean and hornblende facies' metamorphic complex of the Paleoproterozoic. The Middle Proterozoic strata are dominated by sedimentary construction within the relatively stable Craton, lacking obvious magmatic activities, and the lithologies contain yellow-brown conglomerate, dark-gray siltstone, and carbonates. The Neoproterozoic rocks comprise mainly coastal sedimentary rocks and lagoonal evaporites, which belong to an aulacogen sedimentary assemblage [36]. The Lower Paleozoic strata are composed of shallow marine sedimentary rocks and shales of the Cambrian and Lower–Middle Ordovician. The Carboniferous–Permian rocks mainly consist of continental clastic rocks with minor marine clastic rocks [37]. The Mesozoic rocks are widely distributed in this area, including the Lower–Middle Jurassic volcaniclastic rocks and the Upper Jurassic–Lower Cretaceous intermediate-acid volcanic rocks, and the lithologies include brick-red tuff breccia, gray-green andesite, and light flesh-red rhyolite [38]. The Paleocene–Neoproterozoic rocks are composed of clastic and iddingsite basalts.

The northern margin of the NCC is at the junction of the North China, Pacific, and Siberia plates and is characterized by a complex geotectonic evolution. Before the Late Paleozoic, this region was relatively stable with weak tectonic activities. The Late Paleozoic structures were influenced by the tectonic evolution of the Central Asian Orogenic Belt, which resulted in the formation of EW-, NW-, and NE-trending faults (Figure 1b). In the Mesozoic, this region may have experienced the tectonic evolution of collisional orogenesis and extensional orogenesis, and the collisional orogenesis is closely related to the closure of the Paleo-Asian Ocean (PAO) and the collision of continental plates [39], which forms a large-scale, EW-trending Yinshan–Yanshan fold-thrust belt (Figure 1a). Extensional orogenesis is related to the subduction of the Paleo-Pacific plate, and, during this period, this area underwent significant thinning and catastrophic destruction, subsequently developing

a large number of NNE- and NE-trending extensional tectonics [40]. Regional magmatism in the NCC can be divided into the Archean, Proterozoic, Permian, and Mesozoic periods. The Archean intrusive rocks mainly consist of intermediate-acid volcanic rocks and the lithologies contain mainly granodiorite and diorite [41]. The Meso-Neoproterozoic intrusive rocks are mainly composed of plagiogranite and granite, accompanied by some mafic dyke swarms [42]. Since the Phanerozoic, the magmatic activities in this area have been frequent and characterized by multiple phases and a wide distribution. The Permian intrusive rocks are mainly composed of quartz diorite, granodiorite, diorite, and granite. The Mesozoic intrusions are represented by the Triassic alkaline granite and the Jurassic–Cretaceous anorthosite, diorite, granodiorite, and granite porphyry [43,44].

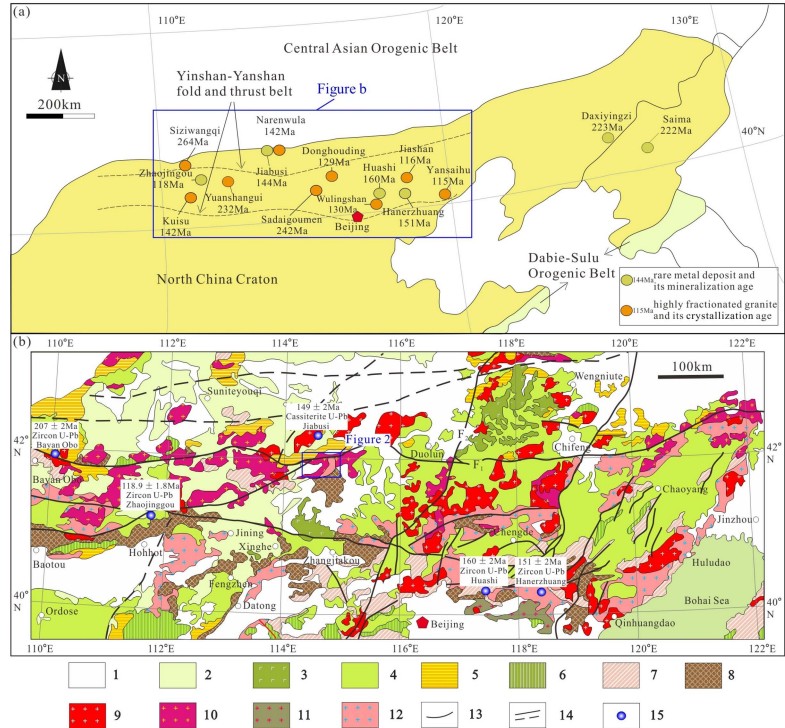

**Figure 1.** The geotectonic map ((**a**) modified after [45]) and geological sketch map ((**b**) modified after [46]) of the northern margin of the NCC. F$_1$ = Chifeng–Kaiyuan fault; F$_2$ = Daxing'anling–Taihangshan fault; 1 = Quaternary sediments; 2 = Paleogene and Neogene sediments; 3 = Neogene basalt; 4 = Mesozoic volcanic rock and clastic rock; 5 = Late Paleozoic strata; 6 = Early Paleozoic strata; 7 = Meso-Neoproterozoic sediments; 8 = Archean-Paleoproterozoic crystalline basement; 9 = Mesozoic granitoids; 10 = Paleozoic granitoids; 11 = Meso-Neoproterozoic granitoids; 12 = Archean-Paleoproterozoic granitoids; 13 = geological boundary; 14 = observed/inferred fault; 15 = rare metal deposit.

## 3. Petrography

The study region is located in Kangbao County, Zhangjiakou City, Hebei Province, which belongs to the MNNCC (Figure 1a). The exposed rocks in the study area mainly include a set of intermediate volcanic rocks of the Lower Permian Elitu Formation, a gray-green, medium-thick layer of tuffaceous sandstones, conglomerates and the dark gray clay shales of the Lower Permian Yujiabeigou Formation, acid and the intermediate acid volcaniclastic rocks of the Upper Jurassic Zhangjiakou Formation, and the purple-gray, unequal-grained conglomerates of the Lower Cretaceous Damoguaihe Formation. In addition, the Quaternary sediments are widely distributed in the study region with a relatively thin thickness, and they contain a large number of plant roots and bioclastic fragments (Figure 2). The study region is influenced by the Indosinian and Yanshanian tectono-magmatic activities, which form the NW- and NE-trending faults, with the former

being large in scale. The periods of magmatism in the study region can be divided into Indosinian and Yanshanian. The Indosinian intrusions are mainly spread in the EW direction, and the lithologies mainly include the KG and FAG, which are mainly produced along the uplift area between the deep faults. The Yanshanian intrusions are mainly spread in the NE and NW directions, with the lithology mainly including K-feldspar granite porphyry.

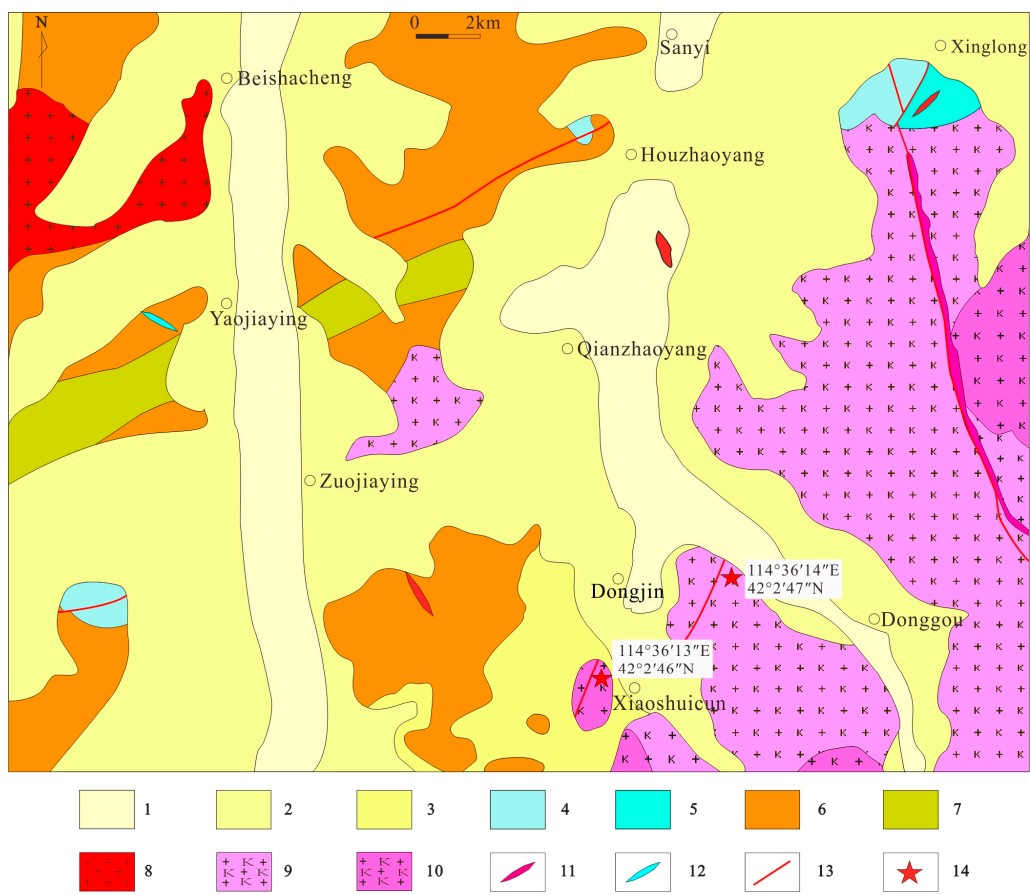

**Figure 2.** Geological sketch map of the study region: 1 = Quaternary Holocene; 2 = Quaternary Pleistocene Malan Formation; 3 = Quaternary Pleistocene Chicheng Formation; 4 = Lower Cretaceous Damoguaihe Formation; 5 = Upper Jurassic Zhangjiakou Formation; 6 = Lower Permian Yujiabeigou Formation; 7 = Lower Permian Elitu Formation; 8 = Yanshanian kali-feldspar granite porphyry; 9 = Indosinian kali-feldspar granite; 10 = Indosinian fine-grained alkali-feldspar granite; 11 = Granite aplite vein; 12 = Quartz vein; 13 = Fault; 14 = Sampling point.

The Dongjin intrusion, located in the middle part of the study region, has intruded into the Lower Permian Yujia'beigou and Elitu Formations as a stock. The FAG and KG within the Dongjin intrusion exhibit vertical zoning, with a gradual transition from shallow to deep zones and no discernible intrusion boundary. Additionally, a small amount of pegmatite is visible at the top of the FAG (Figure 3c). The Rb-Nb-Ta mineralization is mainly developed at the top of the Dongjin intrusion and occurs in the stratoid shape. The alterations that occurred in the mineralized bodies were albitization, amazonitization, and muscovitization, with columbite disseminated in the hosted rocks.

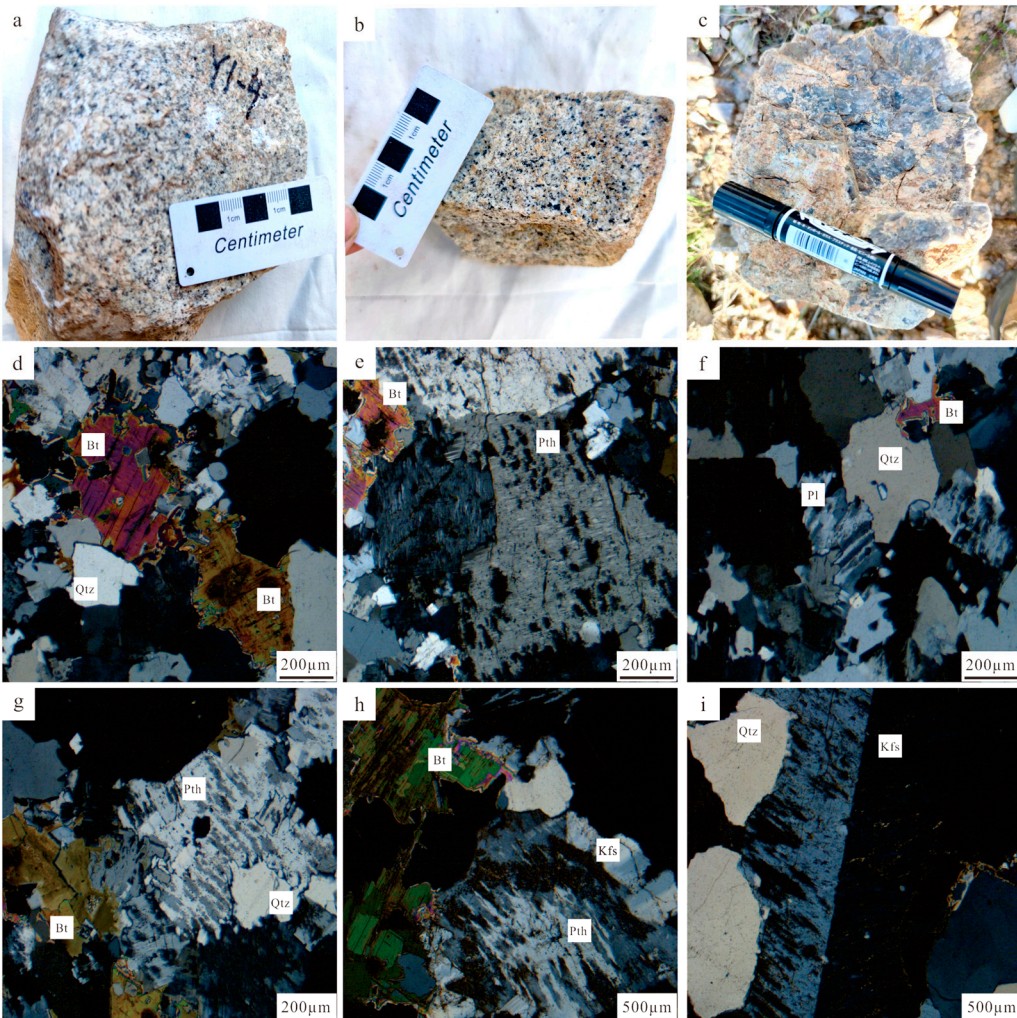

**Figure 3.** Hand specimens and photomicrographs of the Dongjin intrusion: (**a**) hand specimen of the FAG, mainly consisting of K-feldspar, plagioclase, and quartz, followed by biotite; (**b**) hand specimens of the KG, mainly consisting of K-feldspar, plagioclase, and quartz, followed by biotite; (**c**) hand specimen of the pegmatite; (**d**) typical fine-grained granitic texture in the FAG; (**e**) the stripes in some of the perthite in the FAG are irregular; (**f**) the polysynthetic twin is developed in the FAG with slight sericitization; (**g**) anhedral quartz is enclosed by perthite in the FAG; (**h**) typical granitic texture in the KG, K-feldspar has a small crystal size and the Carlsbad twin can be seen clearly; (**i**) the Carlsbad twin of the K-feldspar is developed in the KG and cracks are developed on the quartz surface. Bt = biotite; Kfs = K-feldspar; Pl = plagioclase; Pth = perthite; Qtz = quartz. (**d–i**) under cross-polarized light.

The Dongjin intrusion can be divided into the FAG and KG. The FAG is yellowish-brown to dark grey in color and has a fine-grained granitic texture and massive structure. It mainly comprises subhedral to anhedral K-feldspar (55%–65% of the rock mass), subhedral to anhedral plagioclase (3%–6%), anhedral quartz (20%–35%), and biotite (4%–8%), with minor accessory minerals (~2%, including apatite and zircon). The grain size ranges from 0.2 to 1.0 mm (averaging 0.5 mm, Figure 3d–g). The KG is flesh-red in color and has a granitic texture and massive structure. It is mainly composed of subhedral K-feldspar (40%–50%), euhedral to subhedral plagioclase (4%–8%), anhedral quartz (25%–30%), and subhedral biotite (3%–5%), with minor accessory minerals (~2%, including apatite and zircon). The grain size ranges from 1.0 to 3.5 mm (averaging 2.0 mm, Figure 3h,i). The characteristic Al-rich minerals such as garnet and cordierite were not found in the Dongjin intrusion. Moreover, slight sericitization and argillization occur in both the FAG and KG.

## 4. Sampling and Analytical Methods

### 4.1. Sampling

In this study, we selected 14 samples, including seven FAG samples (Y1-1–Y1-7) and seven KG samples (Y2-1–Y2-7), from the Dongjin intrusion for U-Pb dating, major and trace element analyses, and electron probe microanalysis (EPMA).

Two columbite samples (columbite crystals separated from Y1-4 and Y2-7 samples) were used for laser ablation inductively coupled plasma mass spectrometry (LA-ICP-MS) U-Pb dating (Figure 2). Fourteen samples were collected for major and trace element analyses. One KG sample (Y2-1) was selected for the EPMA of mica and feldspar, respectively.

### 4.2. Columbite U-Pb Dating

The separation of columbite was performed at the Tuoxuan Rock and Mineral Testing Service Co., Ltd., Langfang, Hebei Province, China (TRMTS). After separating two columbite samples using the conventional heavy liquid and magnetic techniques, columbite grains with better crystalline shape, larger size, and fewer cracks were selected under a binocular microscope. The columbite target preparation was completed at the Beijing Yandu Zhongshi Test Technology Co., Ltd., Beijing, China. Firstly, the selected columbite samples were pasted on the slides with double-sided adhesive tape, and a PVC ring was placed on top of it. Then, the curing agent and the epoxy resin were mixed thoroughly and injected into the PVC ring. After that, the sample holders were stripped off from the slides after the resin was completely cured after a period of time of static time. Finally, the target was polished. At the same time, polarized light, reflected light, back-scattered electron imaging (BSE), and LA-ICP-MS columbite U-Pb analysis were also performed. An Analytik Jena Plasma Quant MC-ICP-MS (Analytik Jena Gmbh, Jena, Germany) with an NWR 193 nm Ar-F excimer laser was used for the columbite U-Pb analysis. For the test, blank gas was passed for 15 s followed by 40 s of laser stripping analysis with laser beam spot diameters of 38 μm and 27 μm, a frequency of 6 Hz, and an energy density of 4.0 J/cm$^2$ [47–51]. The raw data were calibrated offline using ICPMSDataCal (China University of Geosciences, Wuhan, China) and Zskits (ZSkits 1.1.0, Yanduzhongshi Geological Analysis Laboratory Ltd., Beijing, China), and the common Pb correction was performed using the $^{207}$Pb method [52,53]. The calculations of the U-Pb age and the drawing of concordia diagrams were performed with Isoplot software [54]. The lower intercept represents the age of columbite in the Tera–Wasserburg diagram.

### 4.3. Major and Trace Element Analyses

Major and trace element compositions were measured at the Beijing Yandu Zhongshi Test Technology Co., Ltd., Beijing, China. Major element compositions were analyzed using a Leeman Prodigy inductively coupled plasma-optical emission spectrometry (ICP-OES) system (LEEMAN LABS INC, Hudson, NH, USA) with a high-dispersion Echelle optics system. The powdered samples were first mixed with anhydrous lithium metaborate and heated to 1000 °C for full fusion; then, the fused sample was left to cool down to room temperature. After cooling, distilled water containing HNO$_3$ was added to the solution and stirred to dissolve it until a clear and stable solution was formed. Finally, these solutions were diluted for ICP-OES analysis. The detailed procedure of the analysis and the detailed parameters of the instrument were mentioned by Thompson and Walsh [55]. The analytical errors were analyzed using the US Geological Survey rock standards BCR-1 and AVG-2, as well as the Chinese national rock standards GSR-3, in which the analytical accuracies of TiO$_2$ and P$_2$O$_5$ were around 1.5% and 2.0%, respectively, and the analytical accuracies of other oxides were better than 1%. Agilent-7500a inductively coupled plasma mass spectrometry (ICP-MS) was used for the analysis and determination of rare earth elements and trace elements. Firstly, 40 mg of powder samples were thoroughly mixed with 1.0 mL of HF and 0.5 mL of HNO$_3$ in high-pressure PTFE bombs. Secondly, these bombs were placed in an oven at 195 °C for 72 h. Finally, rock digestion diluent was nebulized into the Agilent-7500a ICP-MS (Agilent Technologies, Santa Clara, CA, USA) for the determination

of trace elements. The reference materials BCR-1 and BHVO-1 from the US Geological Survey were utilized to monitor the quality of the data, and the majority of the rare earth and trace elements were analyzed with a precision of more than 5%.

### 4.4. Electron Probe Microanalysis

The EPMA of mica and feldspar minerals was carried out at the Institute of Mineral Resources, Chinese Academy of Sciences (CAGS), using a JEOL JXA-8230 (JEOL, Ltd, Tokyo, Japan) electron micro-probe instrument, five wavelength dispersive spectrometers (WDSs) for quantitative analysis, and one energy-dispersive spectrometer (EDS) for qualitative analysis, where the specific operating conditions were as follows: accelerating voltage of 15 kV, beam current of 10 nA, and beam spot of 5 um. The background counting time was 30 s and the peak counting time was 60 s for the analysis of F, Cl, and Ti; moreover, a 15 s background time and a 30 s peak counting time were used for the analysis of the other elements. The use of natural minerals and synthetic oxides for calibration includes wollastonite (Si), anorthite (Al and Ca), rutile (Ti), MnO (Mn), hematite (Fe), jadeite (Na), K-feldspar (K), olivine (Mg), apatite (P), $Cr_2O_3$ (Cr), nickelite (Ni), topaz (F), and NaCl (Cl). All data were calibrated using the ZAF 164 procedure, with the specific requirement that the limit of detection be 0.01% and that analytical measurements be subjected to relative uncertainties, which were 1% for major elements and 4% for minor elements.

## 5. Analytical Results

### 5.1. Columbite U-Pb Age

A backscattered electron image of columbite shows that the crystal composition of columbite in the two samples tested (Y1-4 and Y2-7) is relatively homogeneous, structurally simple (no zoning structure is seen), and mostly conical (Figure 4). Columbite U-Pb results are listed in Table 1. Most of the analytical points measured for the two samples are located on the concordant line (Figure 5). A total of 30 spots for sample Y1-4 gave a lower intercept age of 248.9 ± 1.9 Ma (*n* = 30, MSWD = 1.9) on the Tera–Wasserburg concordia diagram (Figure 5a); 30 spots for sample Y2-7 gave a lower intercept age of 248.9 ± 1.9 Ma (*n* = 30, MSWD = 0.99; Figure 5b). The two obtained ages are consistent, thereby suggesting that the Dongjin intrusion and Nb-Ta mineralization were formed in the Early Triassic.

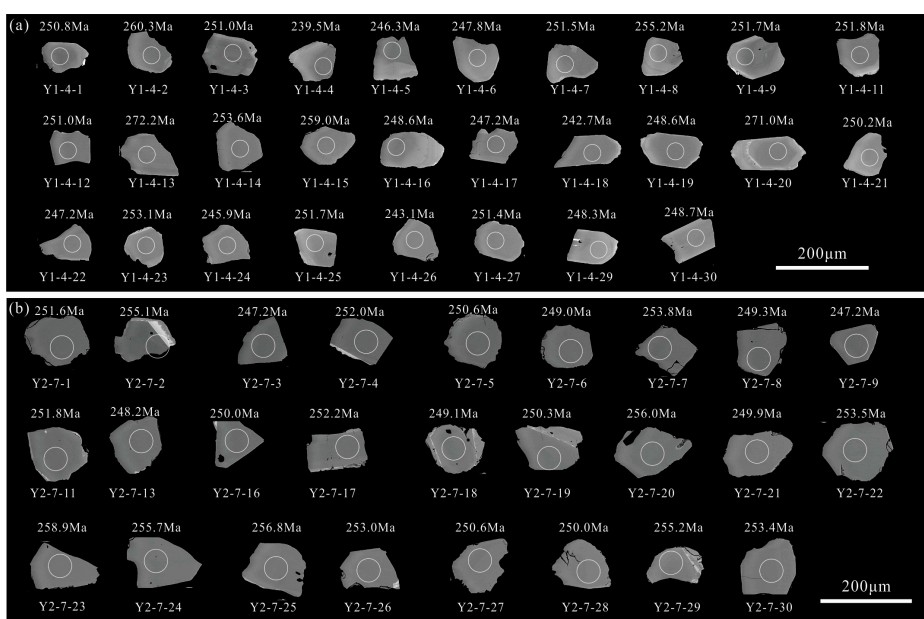

**Figure 4.** Backscattered electron image of columbite in the FAG (**a**) and KG (**b**) from the Dongjin intrusion.

**Table 1.** LA-ICP-MS columbite U-Pb data for the FAG and KG (samples Y1-4 and Y2-7) from the Dongjin intrusion.

| Spot No. | Th (ppm) | U (ppm) | U/Th | Isotopic Ratios | | | | | | | | Ages (Ma) | | | |
|---|---|---|---|---|---|---|---|---|---|---|---|---|---|---|---|
| | | | | $^{207}Pb/^{235}U$ | $1\sigma$ | $^{206}Pb/^{238}U$ | $1\sigma$ | $^{206}Pb/^{238}U$ | $1\sigma$ | $^{207}Pb/^{206}Pb$ | $1\sigma$ | $^{207}Pb/^{235}U$ | $1\sigma$ | $^{206}Pb/^{238}U$ | $1\sigma$ |
| Y1-4-01 | 8.40 | 330 | 39.5 | 0.28 | 0.007 | 0.04 | 0.0007 | 0.05 | 0.001 | 0.05 | 0.001 | 250.8 | 5.6 | 250.8 | 4.3 |
| Y1-4-02 | 35.0 | 687 | 19.6 | 0.29 | 0.007 | 0.04 | 0.0008 | 0.05 | 0.001 | 0.05 | 0.001 | 258.3 | 5.3 | 260.3 | 4.7 |
| Y1-4-03 | 4.20 | 303 | 71.6 | 0.28 | 0.006 | 0.04 | 0.0005 | 0.05 | 0.001 | 0.05 | 0.001 | 253.7 | 4.8 | 251.0 | 3.2 |
| Y1-4-04 | 37.1 | 611 | 16.5 | 0.27 | 0.005 | 0.04 | 0.0004 | 0.05 | 0.001 | 0.05 | 0.001 | 247.0 | 3.7 | 239.5 | 2.4 |
| Y1-4-05 | 10.0 | 284 | 28.5 | 0.28 | 0.007 | 0.04 | 0.0004 | 0.05 | 0.001 | 0.05 | 0.001 | 254.3 | 5.2 | 246.3 | 2.2 |
| Y1-4-06 | 2.60 | 276 | 107 | 0.28 | 0.007 | 0.04 | 0.0006 | 0.05 | 0.001 | 0.05 | 0.001 | 248.2 | 5.2 | 247.8 | 3.6 |
| Y1-4-07 | 4.60 | 238 | 51.3 | 0.29 | 0.007 | 0.04 | 0.0008 | 0.05 | 0.001 | 0.05 | 0.001 | 256.2 | 5.9 | 251.5 | 4.7 |
| Y1-4-08 | 38.8 | 665 | 17.1 | 0.30 | 0.006 | 0.04 | 0.0006 | 0.05 | 0.001 | 0.05 | 0.001 | 266.0 | 4.3 | 255.2 | 3.6 |
| Y1-4-09 | 3.70 | 337 | 90.6 | 0.30 | 0.006 | 0.04 | 0.0006 | 0.06 | 0.001 | 0.06 | 0.001 | 269.9 | 4.8 | 251.7 | 3.6 |
| Y1-4-11 | 4.90 | 230 | 47.3 | 0.30 | 0.009 | 0.04 | 0.0006 | 0.06 | 0.001 | 0.06 | 0.001 | 267.7 | 6.7 | 251.8 | 4.0 |
| Y1-4-12 | 47.6 | 766 | 16.1 | 0.28 | 0.005 | 0.04 | 0.0006 | 0.05 | 0.001 | 0.05 | 0.001 | 254.1 | 4.0 | 251.0 | 3.7 |
| Y1-4-13 | 12.9 | 279 | 21.6 | 0.53 | 0.021 | 0.04 | 0.0008 | 0.09 | 0.003 | 0.09 | 0.003 | 432.0 | 14 | 272.2 | 4.8 |
| Y1-4-14 | 24.3 | 494 | 20.3 | 0.29 | 0.006 | 0.04 | 0.0006 | 0.05 | 0.001 | 0.05 | 0.001 | 257.0 | 4.8 | 253.6 | 3.7 |
| Y1-4-15 | 49.8 | 811 | 16.3 | 0.29 | 0.007 | 0.04 | 0.0008 | 0.05 | 0.001 | 0.05 | 0.001 | 261.7 | 5.4 | 259.0 | 4.8 |
| Y1-4-16 | 8.60 | 277 | 32.2 | 0.28 | 0.007 | 0.04 | 0.0006 | 0.05 | 0.001 | 0.05 | 0.001 | 252.3 | 5.7 | 248.6 | 3.8 |
| Y1-4-17 | 6.30 | 248 | 39.2 | 0.28 | 0.008 | 0.04 | 0.0008 | 0.05 | 0.001 | 0.05 | 0.001 | 252.9 | 6.2 | 247.2 | 4.7 |
| Y1-4-18 | 6.20 | 281 | 45.2 | 0.80 | 0.005 | 0.04 | 0.0004 | 0.05 | 0.001 | 0.05 | 0.001 | 247.9 | 3.9 | 242.7 | 2.7 |
| Y1-4-19 | 5.10 | 234 | 45.7 | 0.28 | 0.007 | 0.04 | 0.0006 | 0.05 | 0.001 | 0.05 | 0.001 | 252.4 | 5.4 | 248.6 | 4.0 |
| Y1-4-20 | 6.40 | 243 | 38.2 | 0.65 | 0.020 | 0.04 | 0.0010 | 0.11 | 0.002 | 0.11 | 0.002 | 507.1 | 12 | 271.0 | 6.4 |
| Y1-4-21 | 30.1 | 516 | 17.1 | 0.28 | 0.007 | 0.04 | 0.0009 | 0.05 | 0.001 | 0.05 | 0.001 | 248.1 | 5.7 | 250.2 | 5.5 |
| Y1-4-22 | 5.30 | 233 | 43.7 | 0.28 | 0.005 | 0.04 | 0.0005 | 0.05 | 0.001 | 0.05 | 0.001 | 248.6 | 4.2 | 247.2 | 3.0 |
| Y1-4-23 | 9.20 | 280 | 30.5 | 0.29 | 0.006 | 0.04 | 0.0005 | 0.05 | 0.001 | 0.05 | 0.001 | 257.6 | 4.3 | 253.1 | 3.3 |
| Y1-4-24 | 5.10 | 280 | 54.8 | 0.28 | 0.007 | 0.04 | 0.0008 | 0.05 | 0.001 | 0.05 | 0.001 | 248.3 | 5.9 | 245.9 | 4.7 |
| Y1-4-25 | 7.80 | 257 | 33.1 | 0.28 | 0.007 | 0.04 | 0.0006 | 0.05 | 0.001 | 0.05 | 0.001 | 254.9 | 5.7 | 251.7 | 3.6 |
| Y1-4-26 | 5.20 | 286 | 55.4 | 0.27 | 0.007 | 0.04 | 0.0007 | 0.05 | 0.001 | 0.05 | 0.001 | 242.1 | 5.9 | 243.1 | 4.2 |
| Y1-4-27 | 10.4 | 372 | 35.8 | 0.29 | 0.006 | 0.04 | 0.0006 | 0.05 | 0.001 | 0.05 | 0.001 | 262.5 | 4.8 | 251.4 | 3.8 |
| Y1-4-29 | 3.00 | 277 | 92.0 | 0.29 | 0.005 | 0.04 | 0.0005 | 0.05 | 0.001 | 0.05 | 0.001 | 257.8 | 4.2 | 248.3 | 3.0 |
| Y1-4-30 | 2.40 | 251 | 105 | 0.28 | 0.006 | 0.04 | 0.0005 | 0.05 | 0.001 | 0.05 | 0.001 | 251.9 | 4.4 | 248.7 | 3.1 |
| Y2-7-01 | 16.0 | 424 | 26.5 | 0.28 | 0.005 | 0.04 | 0.0004 | 0.05 | 0.001 | 0.05 | 0.001 | 254.1 | 3.6 | 251.6 | 2.6 |
| Y2-7-02 | 7.40 | 293 | 39.4 | 0.28 | 0.007 | 0.04 | 0.0007 | 0.05 | 0.001 | 0.05 | 0.001 | 252.4 | 5.9 | 255.1 | 4.5 |
| Y2-7-03 | 23.7 | 578 | 24.4 | 0.28 | 0.005 | 0.04 | 0.0005 | 0.05 | 0.001 | 0.05 | 0.001 | 252.1 | 3.7 | 247.2 | 2.8 |
| Y2-7-04 | 20.4 | 437 | 21.4 | 0.28 | 0.005 | 0.04 | 0.0004 | 0.05 | 0.001 | 0.05 | 0.001 | 253.5 | 3.8 | 252.0 | 2.5 |
| Y2-7-05 | 14.7 | 397 | 27.0 | 0.29 | 0.005 | 0.04 | 0.0004 | 0.05 | 0.001 | 0.05 | 0.001 | 257.7 | 3.9 | 250.6 | 2.5 |
| Y2-7-06 | 16.1 | 386 | 24.0 | 0.28 | 0.005 | 0.04 | 0.0004 | 0.05 | 0.001 | 0.05 | 0.001 | 251.3 | 3.9 | 249.0 | 2.4 |
| Y2-7-07 | 17.6 | 396 | 22.5 | 0.29 | 0.005 | 0.04 | 0.0004 | 0.05 | 0.001 | 0.05 | 0.001 | 256.3 | 3.9 | 253.8 | 2.5 |
| Y2-7-08 | 18.2 | 490 | 26.9 | 0.28 | 0.006 | 0.04 | 0.0008 | 0.05 | 0.001 | 0.05 | 0.001 | 250.5 | 4.6 | 249.3 | 4.7 |
| Y2-7-09 | 13.3 | 369 | 27.7 | 0.28 | 0.004 | 0.04 | 0.0004 | 0.05 | 0.001 | 0.05 | 0.001 | 248.3 | 3.5 | 247.2 | 2.2 |

**Table 1.** *Cont.*

| Spot No. | Th (ppm) | U (ppm) | U/Th | Isotopic Ratios | | | | | | | | | | Ages (Ma) | | | |
|---|---|---|---|---|---|---|---|---|---|---|---|---|---|---|---|---|---|
| | | | | $^{207}Pb/^{235}U$ | $1\sigma$ | $^{206}Pb/^{238}U$ | $1\sigma$ | $^{206}Pb/^{238}U$ | $1\sigma$ | $^{207}Pb/^{206}Pb$ | $1\sigma$ | $^{207}Pb/^{235}U$ | $1\sigma$ | $^{206}Pb/^{238}U$ | $1\sigma$ |
| Y2-7-11 | 7.60 | 306 | 40.3 | 0.29 | 0.006 | 0.04 | 0.0004 | 0.05 | 0.001 | 0.05 | 0.001 | 257.8 | 4.8 | 251.8 | 2.6 |
| Y2-7-13 | 15.3 | 439 | 28.7 | 0.30 | 0.007 | 0.04 | 0.0004 | 0.06 | 0.001 | 0.06 | 0.001 | 268.9 | 5.8 | 248.2 | 2.5 |
| Y2-7-16 | 21.2 | 466 | 22.0 | 0.28 | 0.005 | 0.04 | 0.0005 | 0.05 | 0.001 | 0.05 | 0.001 | 253.2 | 3.8 | 250.0 | 3.0 |
| Y2-7-17 | 20.5 | 477 | 23.3 | 0.28 | 0.006 | 0.04 | 0.0005 | 0.05 | 0.001 | 0.05 | 0.001 | 250.5 | 4.6 | 252.2 | 3.0 |
| Y2-7-18 | 10.7 | 327 | 30.6 | 0.31 | 0.007 | 0.04 | 0.0005 | 0.06 | 0.001 | 0.06 | 0.001 | 276.8 | 5.4 | 249.1 | 3.1 |
| Y2-7-19 | 20.3 | 447 | 22.0 | 0.29 | 0.006 | 0.04 | 0.0005 | 0.05 | 0.001 | 0.05 | 0.001 | 255.8 | 4.4 | 250.3 | 2.9 |
| Y2-7-20 | 29.7 | 369 | 12.4 | 0.28 | 0.006 | 0.04 | 0.0005 | 0.05 | 0.001 | 0.05 | 0.001 | 252.4 | 4.8 | 256.0 | 3.2 |
| Y2-7-21 | 37.3 | 620 | 16.6 | 0.28 | 0.005 | 0.04 | 0.0005 | 0.05 | 0.001 | 0.05 | 0.001 | 252.7 | 4.1 | 249.9 | 3.0 |
| Y2-7-22 | 22.9 | 441 | 19.3 | 0.29 | 0.005 | 0.04 | 0.0005 | 0.05 | 0.001 | 0.05 | 0.001 | 259.1 | 4.2 | 253.5 | 3.0 |
| Y2-7-23 | 12.3 | 372 | 30.2 | 0.37 | 0.007 | 0.04 | 0.0005 | 0.07 | 0.001 | 0.07 | 0.001 | 321.8 | 5.4 | 258.9 | 3.1 |
| Y2-7-24 | 6.20 | 330 | 53.6 | 0.29 | 0.006 | 0.04 | 0.0005 | 0.05 | 0.001 | 0.05 | 0.001 | 256.8 | 4.5 | 255.7 | 3.0 |
| Y2-7-25 | 8.10 | 302 | 37.5 | 0.30 | 0.009 | 0.04 | 0.0007 | 0.05 | 0.001 | 0.05 | 0.001 | 264.6 | 6.8 | 256.8 | 4.1 |
| Y2-7-26 | 41.6 | 602 | 14.5 | 0.28 | 0.006 | 0.04 | 0.0005 | 0.05 | 0.001 | 0.05 | 0.001 | 253.9 | 4.3 | 253.0 | 2.9 |
| Y2-7-27 | 40.3 | 727 | 18.0 | 0.31 | 0.007 | 0.04 | 0.0004 | 0.06 | 0.001 | 0.06 | 0.001 | 275.1 | 5.5 | 250.6 | 2.2 |
| Y2-7-28 | 9.70 | 332 | 34.2 | 0.28 | 0.006 | 0.04 | 0.0005 | 0.05 | 0.001 | 0.05 | 0.001 | 253.0 | 4.9 | 250.0 | 3.3 |
| Y2-7-29 | 52.7 | 854 | 16.2 | 0.31 | 0.005 | 0.04 | 0.0005 | 0.06 | 0.001 | 0.06 | 0.001 | 273.8 | 4.0 | 255.2 | 3.0 |
| Y2-7-30 | 5.20 | 377 | 72.5 | 0.30 | 0.007 | 0.04 | 0.0004 | 0.05 | 0.001 | 0.05 | 0.001 | 263.7 | 5.7 | 253.4 | 2.2 |

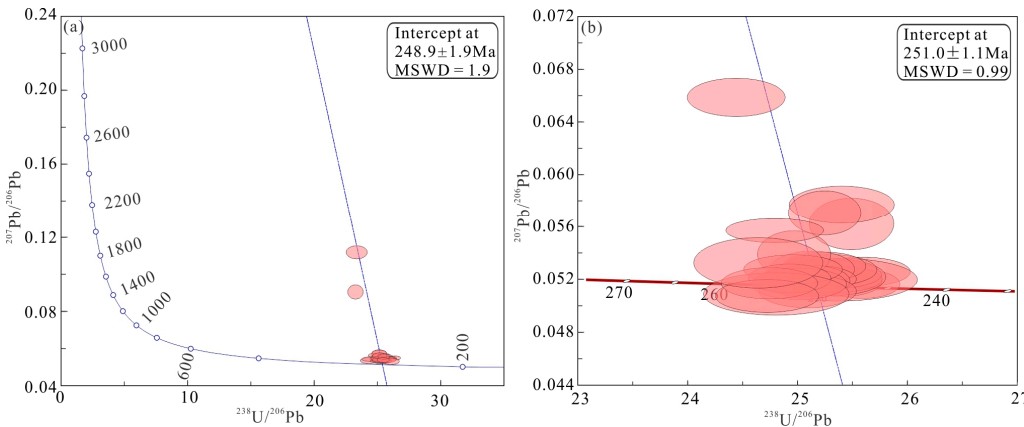

**Figure 5.** U-Pb concordia diagrams (Tera–Wasserburg) of columbite for the FAG (**a**) and KG (**b**) from the Dongjin intrusion.

## 5.2. Whole-Rock Geochemistry

Fourteen whole-rock geochemical data for the FAG and KG samples from the Dongjin intrusion are listed in Table 2.

**Table 2.** Major (wt%) and trace (ppm) element analytical results for the Dongjin intrusion.

| Sample | Y1-1 | Y1-2 | Y1-3 | Y1-4 | Y1-5 | Y1-6 | Y1-7 | Y2-1 | Y2-2 | Y2-3 | Y2-4 | Y2-5 | Y2-6 | Y2-7 |
|---|---|---|---|---|---|---|---|---|---|---|---|---|---|---|
| $SiO_2$ | 76.7 | 76.8 | 76.9 | 77.3 | 77.0 | 77.5 | 76.6 | 77.5 | 77.8 | 77.5 | 77.6 | 77.6 | 77.3 | 77.4 |
| $Al_2O_3$ | 12.6 | 12.8 | 12.9 | 12.4 | 12.6 | 12.4 | 12.6 | 12.3 | 12.1 | 12.0 | 12.0 | 12.0 | 12.1 | 12.1 |
| CaO | 0.16 | 0.13 | 0.16 | 0.23 | 0.19 | 0.21 | 0.24 | 0.31 | 0.40 | 0.33 | 0.30 | 0.29 | 0.33 | 0.34 |
| $K_2O$ | 4.31 | 4.26 | 4.16 | 3.52 | 4.32 | 3.78 | 4.42 | 4.05 | 4.13 | 3.69 | 4.47 | 4.39 | 4.27 | 3.97 |
| $TFe_2O_3$ | 1.20 | 1.17 | 1.08 | 1.23 | 1.15 | 1.19 | 1.24 | 0.98 | 0.71 | 1.40 | 1.20 | 1.17 | 1.21 | 1.25 |
| FeO | 0.90 | 0.84 | 0.59 | 0.79 | 0.84 | 0.84 | 0.84 | 0.46 | 0.27 | 0.65 | 0.36 | 0.67 | 0.42 | 0.75 |
| MgO | 0.05 | 0.07 | 0.05 | 0.04 | 0.06 | 0.05 | 0.07 | 0.09 | 0.09 | 0.09 | 0.07 | 0.08 | 0.09 | 0.09 |
| MnO | 0.05 | 0.05 | 0.05 | 0.05 | 0.05 | 0.05 | 0.05 | 0.04 | 0.04 | 0.04 | 0.04 | 0.05 | 0.04 | 0.05 |
| $Na_2O$ | 4.17 | 4.29 | 4.52 | 4.70 | 4.24 | 4.46 | 4.22 | 4.12 | 3.81 | 4.22 | 3.66 | 3.80 | 3.77 | 4.05 |
| $P_2O_5$ | 0.01 | 0.01 | 0.01 | 0.01 | 0.01 | 0.01 | 0.01 | 0.01 | 0.01 | 0.01 | 0.01 | 0.01 | 0.01 | 0.01 |
| $TiO_2$ | 0.01 | 0.01 | 0.01 | 0.01 | 0.005 | 0.01 | 0.01 | 0.001 | 0.004 | 0.005 | 0.001 | 0.005 | 0.001 | 0.01 |
| LOI | 0.63 | 0.51 | 0.50 | 0.56 | 0.51 | 0.50 | 0.55 | 0.71 | 0.91 | 0.76 | 0.68 | 0.81 | 0.77 | 0.66 |
| Total | 100.0 | 100.1 | 100.4 | 100.0 | 100.1 | 100.2 | 100.0 | 100.1 | 100.0 | 100.0 | 100.0 | 100.2 | 99.9 | 100.0 |
| DI | 96.4 | 96.5 | 97.0 | 96.7 | 95.0 | 96.4 | 96.3 | 96.4 | 96.2 | 95.9 | 96.2 | 96.3 | 95.9 | 95.8 |
| A/NK | 1.10 | 1.10 | 1.08 | 1.07 | 1.17 | 1.09 | 1.07 | 1.11 | 1.13 | 1.09 | 1.10 | 1.09 | 1.12 | 1.11 |
| A/CNK | 1.07 | 1.08 | 1.06 | 1.06 | 1.17 | 1.05 | 1.04 | 1.05 | 1.06 | 1.04 | 1.05 | 1.04 | 1.06 | 1.05 |
| La | 51.8 | 60.3 | 47.4 | 51.0 | 53.7 | 67.6 | 71.3 | 46.2 | 52.1 | 48.5 | 50.1 | 47.8 | 44.8 | 52.4 |
| Ce | 165 | 181 | 152 | 161 | 172 | 201 | 209 | 147 | 170 | 151 | 156 | 146 | 142 | 162 |
| Pr | 14.4 | 16.0 | 13.5 | 15.1 | 15.2 | 15.8 | 17.1 | 12.8 | 13.9 | 13.7 | 14.6 | 13.2 | 13.5 | 15.2 |
| Nd | 41.2 | 47.0 | 38.1 | 43.3 | 41.7 | 44.4 | 54.4 | 38.7 | 40.9 | 41.8 | 47.1 | 39.2 | 41.9 | 48.1 |
| Sm | 9.18 | 10.1 | 8.42 | 10.5 | 9.05 | 8.90 | 12.6 | 8.77 | 8.92 | 10.1 | 11.2 | 8.96 | 10.5 | 12.1 |
| Eu | 0.01 | 0.02 | 0.01 | 0.01 | 0.01 | 0.01 | 0.01 | 0.03 | 0.03 | 0.03 | 0.03 | 0.02 | 0.03 | 0.03 |
| Gd | 6.76 | 7.24 | 6.42 | 7.93 | 6.61 | 6.74 | 9.45 | 7.00 | 6.98 | 8.07 | 8.72 | 7.29 | 8.17 | 9.31 |
| Tb | 1.24 | 1.28 | 1.25 | 1.67 | 1.31 | 1.24 | 1.96 | 1.19 | 1.13 | 1.63 | 1.54 | 1.29 | 1.50 | 1.80 |
| Dy | 7.60 | 7.87 | 8.28 | 11.2 | 8.96 | 8.33 | 12.9 | 7.36 | 6.87 | 10.8 | 9.48 | 8.02 | 9.32 | 10.8 |
| Ho | 1.36 | 1.37 | 1.58 | 2.26 | 1.74 | 1.61 | 2.52 | 1.30 | 1.27 | 2.20 | 1.93 | 1.45 | 1.81 | 2.04 |
| Er | 5.10 | 5.18 | 6.13 | 8.33 | 6.60 | 6.53 | 10.1 | 5.03 | 4.98 | 7.84 | 7.26 | 5.52 | 6.96 | 7.27 |
| Tm | 0.88 | 0.92 | 1.08 | 1.43 | 1.21 | 1.19 | 1.73 | 0.87 | 0.88 | 1.26 | 1.15 | 0.97 | 1.15 | 1.19 |
| Yb | 6.41 | 6.74 | 7.64 | 11.4 | 8.84 | 9.15 | 13.5 | 6.44 | 6.42 | 9.89 | 8.76 | 6.76 | 8.12 | 8.47 |
| Lu | 0.95 | 0.98 | 1.16 | 1.52 | 1.34 | 1.38 | 1.91 | 1.01 | 1.02 | 1.33 | 1.32 | 1.05 | 1.24 | 1.29 |
| Y | 33.7 | 30.1 | 39.4 | 57.2 | 37.8 | 37.2 | 71.9 | 36.8 | 38.4 | 65.3 | 57.7 | 41.7 | 54.2 | 58.7 |
| ΣREE | 312 | 346 | 293 | 326 | 329 | 373 | 418 | 284 | 316 | 308 | 319 | 287 | 291 | 332 |
| LREE | 282 | 314 | 260 | 281 | 292 | 337 | 364 | 254 | 286 | 265 | 279 | 255 | 253 | 289 |
| HREE | 30.3 | 31.6 | 33.5 | 45.8 | 36.6 | 36.2 | 54.1 | 30.2 | 29.5 | 43.0 | 40.2 | 32.4 | 38.3 | 42.1 |
| LREE/HREE | 9.30 | 9.94 | 7.74 | 6.13 | 7.98 | 9.32 | 6.73 | 8.40 | 9.68 | 6.16 | 6.95 | 7.88 | 6.60 | 6.87 |
| $La_N/Yb_N$ | 5.79 | 6.42 | 4.45 | 3.21 | 4.35 | 5.30 | 3.79 | 5.15 | 5.82 | 3.52 | 4.10 | 5.07 | 3.96 | 4.44 |
| Eu/Eu* | 0.004 | 0.01 | 0.004 | 0.005 | 0.004 | 0.005 | 0.003 | 0.01 | 0.01 | 0.01 | 0.01 | 0.01 | 0.01 | 0.01 |
| δCe | 1.48 | 1.43 | 1.48 | 1.42 | 1.48 | 1.51 | 1.46 | 1.49 | 1.55 | 1.44 | 1.42 | 1.42 | 1.41 | 1.40 |
| Rb | 995 | 1019 | 950 | 853 | 912 | 865 | 978 | 711 | 638 | 681 | 761 | 740 | 722 | 760 |
| Ba | 3.92 | 7.46 | 2.69 | 3.80 | 4.90 | 6.42 | 5.27 | 17.1 | 28.8 | 8.41 | 43.9 | 15.1 | 40.9 | 12.6 |
| Th | 50.0 | 48.5 | 43.7 | 49.6 | 50.6 | 55.3 | 59.3 | 53.6 | 65.5 | 55.3 | 52.2 | 49.9 | 50.8 | 60.8 |
| U | 2.32 | 2.93 | 1.82 | 2.40 | 2.18 | 2.63 | 3.13 | 2.41 | 5.10 | 4.37 | 5.00 | 2.35 | 4.95 | 2.96 |
| Nb | 144 | 177 | 131 | 151 | 131 | 165 | 138 | 95.0 | 100 | 128 | 121 | 142 | 82.0 | 139 |
| Ta | 8.79 | 12.2 | 8.92 | 8.79 | 8.19 | 8.92 | 7.78 | 8.79 | 8.52 | 10.1 | 9.11 | 11.2 | 8.04 | 9.69 |
| Sr | 6.38 | 6.07 | 2.63 | 8.02 | 6.15 | 8.18 | 5.18 | 8.11 | 8.41 | 6.09 | 8.73 | 6.85 | 7.84 | 6.38 |
| Zr | 121 | 129 | 99.0 | 120 | 117 | 191 | 251 | 114 | 160 | 138 | 134 | 129 | 125 | 144 |
| Hf | 9.60 | 10.4 | 8.18 | 9.87 | 10.5 | 12.5 | 15.1 | 8.17 | 10.2 | 9.36 | 8.67 | 8.40 | 7.98 | 10.3 |

Seven samples of the FAG from the Dongjin intrusion are characterized by high $SiO_2$ (76.60%–77.50%) and total alkali contents ($Na_2O + K_2O$ = 8.22%–8.68%), medium to low $Al_2O_3$ contents (12.41%–12.94%), poor CaO (0.13%–0.24%), $Fe_2O_3^T$ (1.08%–1.24%), FeO (0.594%–0.901%), MgO (0.04%–0.07%), MnO (0.048%–0.051%), and $TiO_2$ (0.005%–0.01%). In the QAP classification (Figure 6a), most samples plot in the alkali feldspar granite field. In the $R_1$ versus $R_2$ diagram, all the samples fall into the alkaline granite field (Figure 6b). In the $SiO_2$ versus $K_2O$ diagram, all the samples are classified as high-K calc-alkaline (Figure 6c). The samples with FeO/(FeO + MgO) values ranging from 0.92 to 0.95 belong

to the ferroan granitoid. At the same time, the FAG has a high differentiation index (DI = 95.0–97.0). In the $Al_2O_3/(CaO + Na_2O + K_2O)$ (A/CNK) versus $Al_2O_3/(Na_2O + K_2O)$ (A/NK) diagram, all samples are peraluminous (Figure 6d). In the chondrite-normalized diagram (Figure 7a), the REE patterns lean to the right, appear as a seagull, and have total REE contents of 293–418 ppm and relatively strong fractionation between light rare earth elements (LREEs) and heavy rare earth elements (HREEs) (6.13–9.94, averaging 8.16), with $La_N/Yb_N$ values varying from 3.21 to 6.42 (averaging 4.76). They are characterized by extremely negative Eu anomalies (Eu/Eu* = 0.003–0.008, averaging 0.005) with a noticeable tetrad effect. A primitive mantle-normalized, multi-element diagram indicates that these rocks are enriched in Th, U, Nb, Zr, Y, and Rb and depleted in Ba, Sr, P, and Ti (Figure 7b).

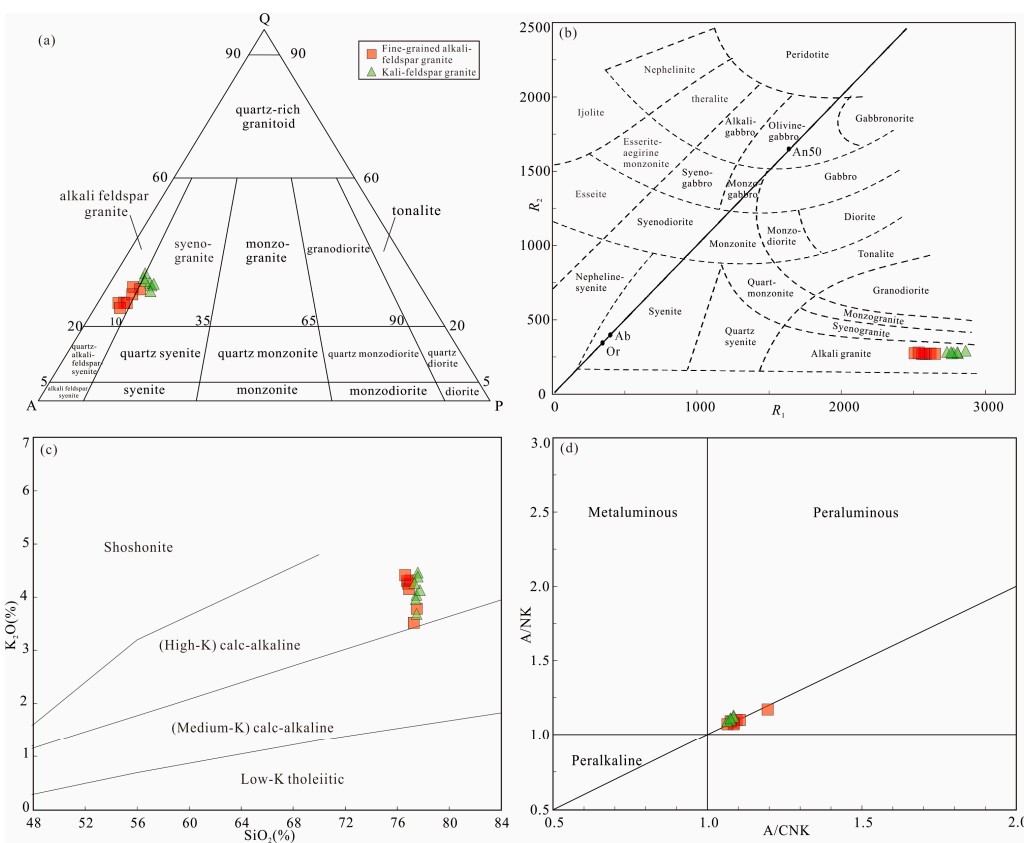

**Figure 6.** Diagrams of petrographic classification for the Dongjin intrusion: (**a**) general classification and nomenclature of plutonic rocks according to mineral content (in vol.%) [56]; Q = quartz, A = alkali feldspar, P = plagioclase; (**b**) $R_1$ versus $R_2$ diagram [57]; (**c**) $SiO_2$ versus $K_2O$ diagram [58]; (**d**) A/CNK versus A/NK diagram [59].

The KG has high $SiO_2$ (77.32%–77.75%, averaging 77.52%) and total alkali content ($Na_2O + K_2O$ = 7.94%–8.19%). The $Al_2O_3$ content (11.96%–12.34%) is lower than that of the FAG. The A/NCK ratios are between 1.04 and 1.06. In the QAP classification (Figure 6a), most samples are plotted in the syenogranite field. Like the FAG, it has the characteristics of being high in silicon and rich in alkali, belonging to the high-K calc-alkaline series. It has poor CaO (0.29%–0.40%), $Fe_2O_3^T$ (0.71%–1.40%), FeO (0.268%–0.747%), MgO (0.07%–0.09%), MnO (0.035%–0.048%), and $TiO_2$ (0.001%–0.01%), and the vast majority of the samples with FeO/(FeO + MgO) values varying from 0.74 to 0.90 belong to the ferroan granitoid. The KG has total REE contents of 284–331 ppm, and they are characterized by extremely negative Eu anomalies (Eu/Eu* = 0.009–0.012, averaging 0.01) with a noticeable tetrad effect; however, it is not as obvious as that of the FAG (Figure 7a). In the primitive mantle-normalized, multi-element diagram (Figure 7b), all samples show obvious enrichments in Th, U, Nb, Zr, Y, and Rb and weak depletions in Ba, Sr, P, and Ti.

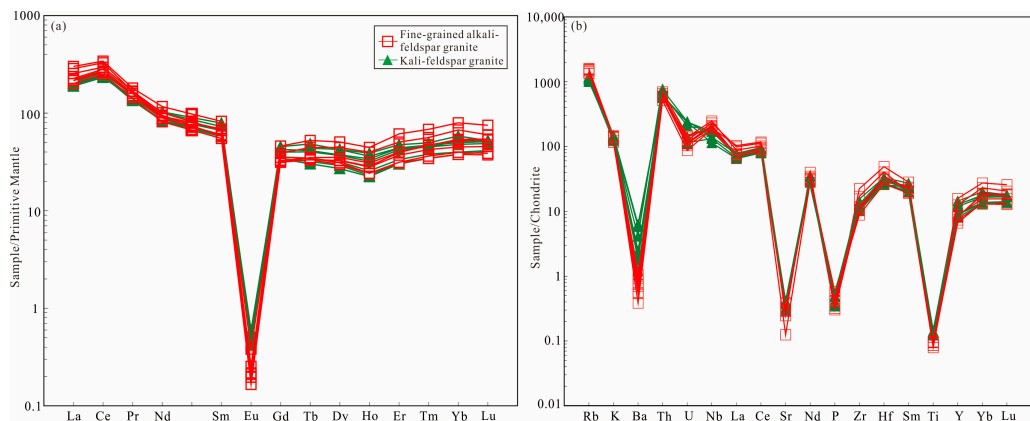

**Figure 7.** Chondrite-normalized REE patterns ((**a**), normalization values after Boynton [60]) and primitive mantle-normalized trace element spider diagrams ((**b**), normalization values after Sun and McDonough [61]) of the Dongjin intrusion.

### 5.3. EPMA of Mica and Feldspar

The analyzed micas and feldspars with the exact sites of the spot analyses are shown in Figure 8. The EPMA data of the mica of the KG samples are shown in Table 3, and the EPMA data of the mica samples are listed in Table 4. The mica samples are characterized by a high content of elements such as Al, Li, K, and F and low content of elements such as Ti, Mn, Mg, Cl, and Fe, and the plagioclase samples have very low An (0.17–1.97) and Or (0.41–3.27) values and extremely high Ab values, varying from 94.8 to 99.2.

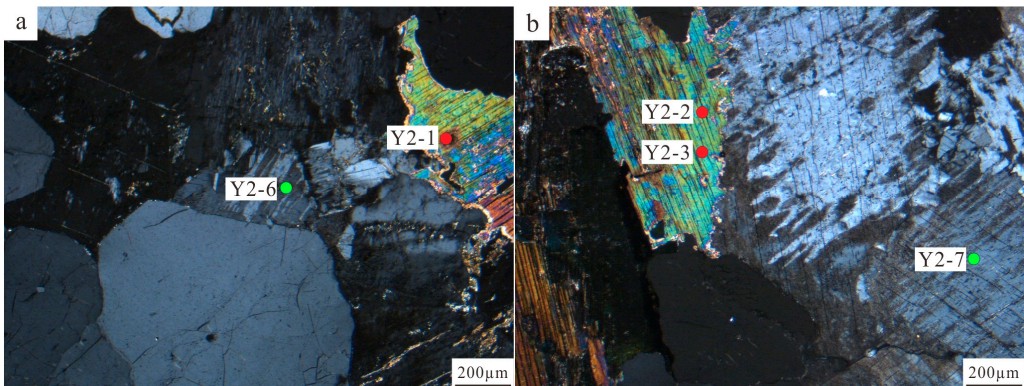

**Figure 8.** Representative photomicrographs of the Dongjin intrusion, showing EMPA analysis spots of anhedral K-feldspar and mica (**a**), alongside subhedral K-feldspar and mica (**b**).

**Table 3.** The electron probe microanalysis (EPMA) data for mica samples (wt%).

| Sample | Y2-1 | Y2-2 | Y2-3 | Y2-4 | Y2-5 |
|---|---|---|---|---|---|
| $SiO_2$ | 39.9 | 42.5 | 40.3 | 40.4 | 41.3 |
| $TiO_2$ | 0.71 | 0.57 | 0.84 | 0.53 | 0.54 |
| $Al_2O_3$ | 18.7 | 20.0 | 19.6 | 19.5 | 20.0 |
| FeO | 25.0 | 22.2 | 23.8 | 23.2 | 23.3 |
| MnO | 0.03 | 0.10 | 0.10 | 0.04 | 0.10 |
| MgO | 0.07 | 0.08 | 0.07 | 0.07 | 0.09 |
| CaO | 0.01 | 0.01 | 0.01 | 0.06 | 0.01 |
| $Na_2O$ | 0.16 | 0.07 | 0.05 | 0.10 | 0.14 |
| $K_2O$ | 9.32 | 8.70 | 9.17 | 9.49 | 8.99 |
| F | 2.81 | 4.40 | 1.96 | 3.87 | 4.52 |
| Cl | 0.07 | 0.01 | 0.07 | 0.03 | 0.07 |

**Table 3.** *Cont.*

| Sample | Y2-1 | Y2-2 | Y2-3 | Y2-4 | Y2-5 |
|---|---|---|---|---|---|
| Number of cations calculated on the basis of 12 oxygen atoms | | | | | |
| Si | 3.01 | 3.07 | 3.04 | 3.01 | 3.00 |
| $Al^{IV}$ | 0.99 | 0.93 | 0.96 | 0.99 | 1.00 |
| $Al^{VI}$ | 0.68 | 0.76 | 0.79 | 0.71 | 0.71 |
| Ti | 0.04 | 0.03 | 0.05 | 0.03 | 0.03 |
| $Fe^{3+}$ | 0.77 | 1.04 | 0.75 | 0.89 | 0.98 |
| $Fe^{2+}$ | 0.82 | 0.30 | 0.75 | 0.55 | 0.43 |
| Mn | 0.002 | 0.01 | 0.01 | 0.003 | 0.01 |
| Mg | 0.01 | 0.01 | 0.01 | 0.01 | 0.01 |
| Ca | 0.001 | 0.001 | 0.0004 | 0.05 | 0.001 |
| Na | 0.02 | 0.01 | 0.01 | 0.01 | 0.02 |
| K | 0.90 | 0.80 | 0.88 | 0.90 | 0.83 |
| Total | 7.23 | 6.96 | 7.25 | 7.11 | 7.02 |
| $Fe^{t}$ | 1.59 | 1.34 | 1.50 | 1.44 | 1.41 |
| $OH^{-}$ | 2.00 | 2.00 | 2.00 | 2.00 | 2.00 |
| MF | 0.005 | 0.01 | 0.005 | 0.005 | 0.01 |
| $Al^{VI} + Fe^{3+} + Ti$ | 1.49 | 1.84 | 1.59 | 1.63 | 1.72 |
| $Fe^{2+} + Mn$ | 0.82 | 0.30 | 0.76 | 0.55 | 0.44 |
| $Ti/(Mg + Fe + Ti + Mn)$ | 0.02 | 0.02 | 0.03 | 0.02 | 0.02 |
| $Al/(Al + Mg + Fe + Ti + Mn + Si)$ | 0.26 | 0.28 | 0.27 | 0.28 | 0.28 |
| $Li_2O$ | 6.27 | 6.16 | 6.38 | 6.28 | 5.95 |
| Li | 1.92 | 1.89 | 1.96 | 1.93 | 1.83 |
| $Mg - Li$ | $-1.91$ | $-1.88$ | $-1.95$ | $-1.92$ | $-1.82$ |
| $Fe^{t} + Mn + Ti - Al^{IV}$ | 0.94 | 0.61 | 0.77 | 0.76 | 0.74 |

**Table 4.** The electron probe microanalysis (EPMA) data for feldspar samples (wt%).

| Sample | Y2-6 | Y2-7 | Y2-8 | Y2-9 | Y2-10 |
|---|---|---|---|---|---|
| $SiO_2$ | 74.7 | 74.6 | 74.9 | 74.3 | 74.6 |
| $Al_2O_3$ | 19.1 | 18.9 | 18.9 | 18.9 | 18.0 |
| CaO | 0.07 | 0.05 | 0.04 | 0.03 | 0.24 |
| $Na_2O$ | 6.53 | 6.99 | 6.48 | 7.89 | 6.24 |
| $K_2O$ | 0.14 | 0.04 | 0.08 | 0.08 | 0.33 |
| BaO | 0.00 | 0.03 | 0.02 | 0.00 | 0.03 |
| Number of cations calculated on the basis of 8 oxygen atoms | | | | | |
| Si | 3.15 | 3.15 | 3.16 | 3.13 | 3.18 |
| Al | 0.95 | 0.94 | 0.94 | 0.94 | 0.90 |
| Ca | 0.003 | 0.002 | 0.002 | 0.0001 | 0.01 |
| Na | 0.53 | 0.57 | 0.53 | 0.65 | 0.52 |
| K | 0.01 | 0.002 | 0.004 | 0.005 | 0.02 |
| Ba | 0.00 | 0.001 | 0.001 | 0.00 | 0.001 |
| An | 0.56 | 0.36 | 0.36 | 0.17 | 1.97 |
| Ab | 98.1 | 99.2 | 98.8 | 99.1 | 94.8 |
| Or | 1.33 | 0.41 | 0.80 | 0.69 | 3.27 |

## 6. Discussion

### 6.1. Rock- and Ore-Forming Ages

Among previous studies, the crystallization ages of several highly fractionated granites along the northern margin of the NCC were concentrated in the Yanshanian period, and several were formed in the Indosinian and Variscan periods. For example, a zircon U-Pb age of $144.3 \pm 0.7$ Ma for the highly fractionated granites from the Jiabusi Nb-Ta deposit was reported by Zhang et al. [9]. The zircon U-Pb age of the ore-forming granites of the Han'erzhuang Nb-Ta deposit was $151 \pm 2.0$ Ma [16]. Qi et al. [12] determined that the U-Pb ages of two zircon grains from the Madi intrusion of the Hua'shi Rb deposit were

156 ± 2 Ma and 165 ± 2 Ma, respectively. Chen et al. [62] measured the zircon U-Pb ages of 243.7 ± 1.6 to 242.2 ± 2 Ma for the alkali feldspar granite from the Sadaigoumen porphyry Mo deposit. In recent years, several rare metal deposits have been discovered successively in the northern margin of the NCC. Previous studies on the crystallization and metallogenic ages of these deposits have been conducted. Li et al. [7] reported magmatic and hydrothermal zircon LA-ICP-MS U-Pb dating in the Zhaojinggou Ta-Nb polymetallic deposit, and the results demonstrate that the weighted average age of magmatic zircon and hydrothermal zircon from amazonitized granite was 116 ± 2 Ma and 112.8 ± 2.2 Ma, respectively. The cassiterite U-Pb age for highly fractionated granite in the Jiabusi Nb-Ta deposit was 149 ± 2.0 Ma, indicating that the deposit was formed in the Late Jurassic [9]. Zhong et al. [11] carried out biotite $^{40}$Ar-$^{39}$Ar dating for the Saima Niobium deposit and obtained an isochron age of 222 Ma for one aegirine–nepheline syenite sample related to mineralization. Sixteen zircon grains of typical magmatic origin from Saima syenite yielded a U-Pb age of 229.5 ± 2.2 Ma, indicating that the ore-forming aegirine nepheline syenite was formed in the Late Triassic [10]. To sum up, the mineralization was concentrated from the Late Triassic to the Early Cretaceous.

The zircons within the Dongjin intrusion exhibit high U content with significant metamictization. Therefore, despite our efforts in attempting zircon U-Pb dating, we unfortunately failed to obtain reliable ages. Columbite, with high U and low common Pb contents, is hardly affected by late hydrothermal alteration. Therefore, the U-Pb dating of columbite is particularly precise, and convincing ages have been obtained in a multitude of studies [24–29]. In this study, columbite U-Pb dating for the FAG and KG yielded ages of 248.9 ± 1.9 Ma and 250.1 ± 1.1 Ma, respectively, which indicates that the Nb-Ta mineralization occurred in the Early Triassic. In conclusion, the Early Triassic Nb-Ta mineralization related to highly fractionated granites in the MNNCC deserves attention.

### 6.2. Petrogenetic Type

As mentioned above, the FAG and KG are rich in silicon and alkali and poor in Ca, Fe, Mg, Ti, and P. They are classified as a high-K calc-alkaline series, showing extremely negative Eu anomalies with a noticeable tetrad effect. The FAG has more pronounced negative Eu anomalies and stronger depletions of Ba, Sr, P, and Ti than the KG. These geochemical characteristics show that the Dongjin intrusion is an A-type granite affinity [63]. In the 10,000 Ga/Al versus ($K_2O + Na_2O$)/CaO diagram [64], all the samples of the Dongjin intrusion fall into the A-type granite field (Figure 9a). The relatively low abundance of mafic minerals (including the lack of amphibole) is also typical of A-type granites [65]. Additionally, the fundamental feature of A-type granite is the enrichment of Fe compared to Mg [66,67]. Both the FAG and KG belong to alkali-calcic ferroan granite.

Highly fractionated granites have similar geochemical characteristics to A-type granites (e.g., high $SiO_2$ and relatively high alkali contents, low CaO and MgO contents, peraluminous and ferroan characters, a "seagull-type" REE pattern with a deep, negative Eu anomaly, enrichment in U, Th, Zr, Rb, Nb, and Y, and depletion in Ba, Sr, P and Ti), which complicates their differentiation [63,67]. Significant effort has been applied in identifying A-type granites and highly fractionated granites [65–68]. Typical A-type granites are formed in high-temperature environments (averaging 839 °C) with high Zr contents [68]. The Zr content of the Dongjin intrusion (average $140.93 \times 10^{-6}$) is significantly lower than that of A-type granite ($>250 \times 10^{-6}$) [69]. The Zr saturation temperature of the Dongjin intrusion averages 780 °C, which is similar to the average temperature of I-type granite (781 °C) [68]. A-type granite tends to evolve from A-type granite toward highly fractionated granite in the 10,000 Ga/Al versus Zr discriminant diagram (Figure 9b) [69], whereas I-type and S-type granite show the opposite trend of evolution to A-type granite, with an increase in the 10,000 Ga/Al ratio. The Dongjin intrusion shows the same trend as the I/S differentiated granites in the process of differentiation. In the $SiO_2$ versus $P_2O_5$ correlation diagram (Figure 10a), the content of $P_2O_5$ decreases with increased $SiO_2$ content, showing a negative correlation. In the Rb versus Y diagram (Figure 10b) and Rh versus Th diagram

(Figure 10c), the contents of Y and Th increase with an increase in Rb content, which shows a positive correlation and indicates that the Dongjin intrusion adheres to a trend of evolving I-type granite. Petrographic studies show that no aluminum-rich minerals (e.g., garnet and cordierite) are found in the Dongjin intrusion combined with the low $P_2O_5$ content, which excludes the possibility of S-type granite. Above all, the Dongjin intrusion is more likely to be highly differentiated I-type granite. However, A-type granite cannot be entirely excluded as no typical I-type granite minerals like amphibole and sphene have been detected in the Dongjin intrusion. Many granitic intrusions have both I-type and A-type granite affinities, such as Dabaishitou pluton [70] and Fogang batholith [71]. In conclusion, we propose that the Dongjin intrusion is a highly differentiated I-type or A-type granite.

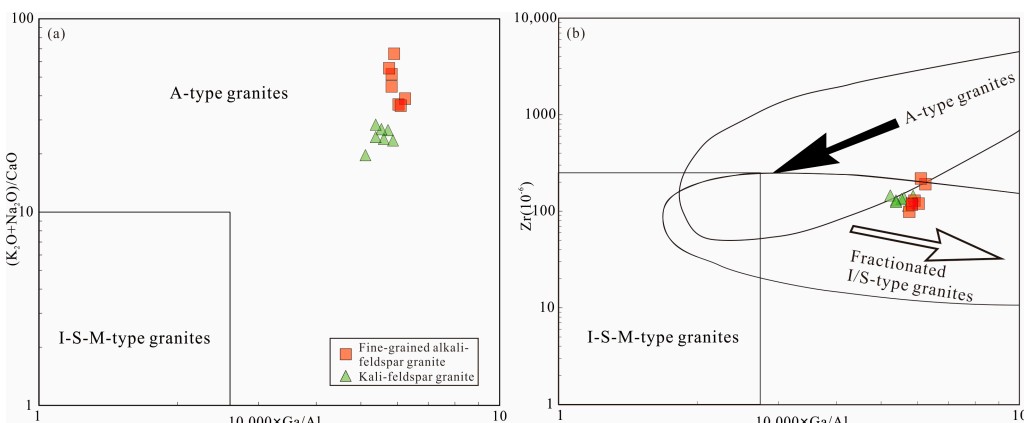

**Figure 9.** $10,000 \times Ga/Al$ versus $(K_2O + Na_2O)/CaO$ (**a**), and $10,000 \times Ga/Al$ versus Zr (**b**) chemical discrimination diagrams for types of granite (after Whalen [64]; the evolution trend line in Figure 9b after [69]).

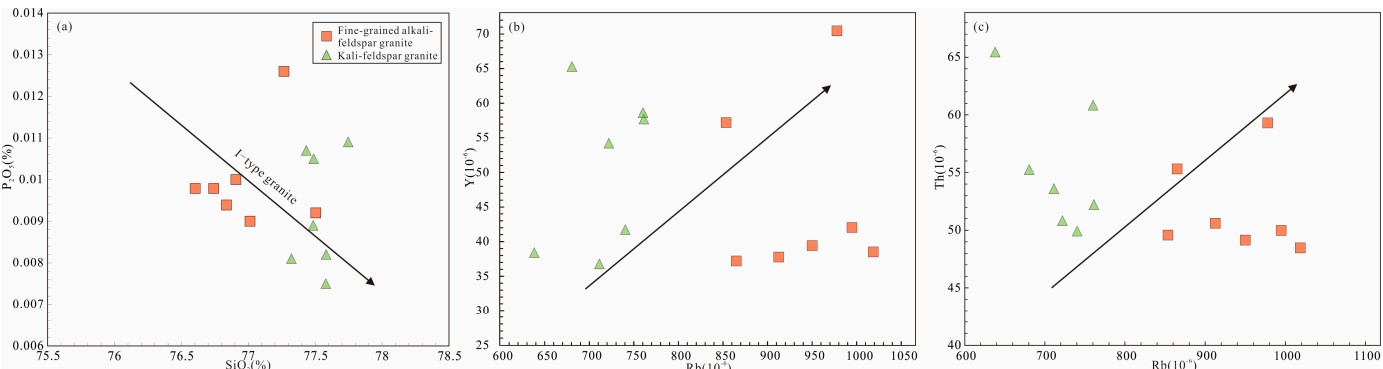

**Figure 10.** $SiO_2$ versus $P_2O_5$ (**a**), Rb versus Y (**b**), and Rb versus Th (**c**) diagrams of the Dongjin intrusion.

### 6.3. Magmatic Evolution and Mineralization

Magmatic evolution is an essential factor controlling the enrichment and mineralization of Nb and Ta in rare metal granites [45,72]. Most Nb-Ta-mineralized granites show peraluminous characteristics, and, in the process of magma evolution, the contents of Nb and Ta are gradually enriched in magma. In the Zr/Hf versus $FeO^T/MgO$ and Zr/Hf versus 10,000 Ga/Al discriminant diagrams, from the KG to the FAG, the $FeO^T/MgO$ and 10,000 Ga/Al ratios increase with a decrease in Zr/Hf ratio for most samples (Figure 11a,b). In the Nb versus 10,000 Ga/Al and Ga versus 10,000 Ga/Al discriminant diagrams, as the Nb and Ga contents increase, the $FeO^T/MgO$ and 10,000 Ga/Al ratios increase (Figure 11c,d), indicating that the initial magma of the Dongjin intrusion has low $FeO^T/MgO$ and Ga/Al ratios and the ratios increase with magmatic evolution.

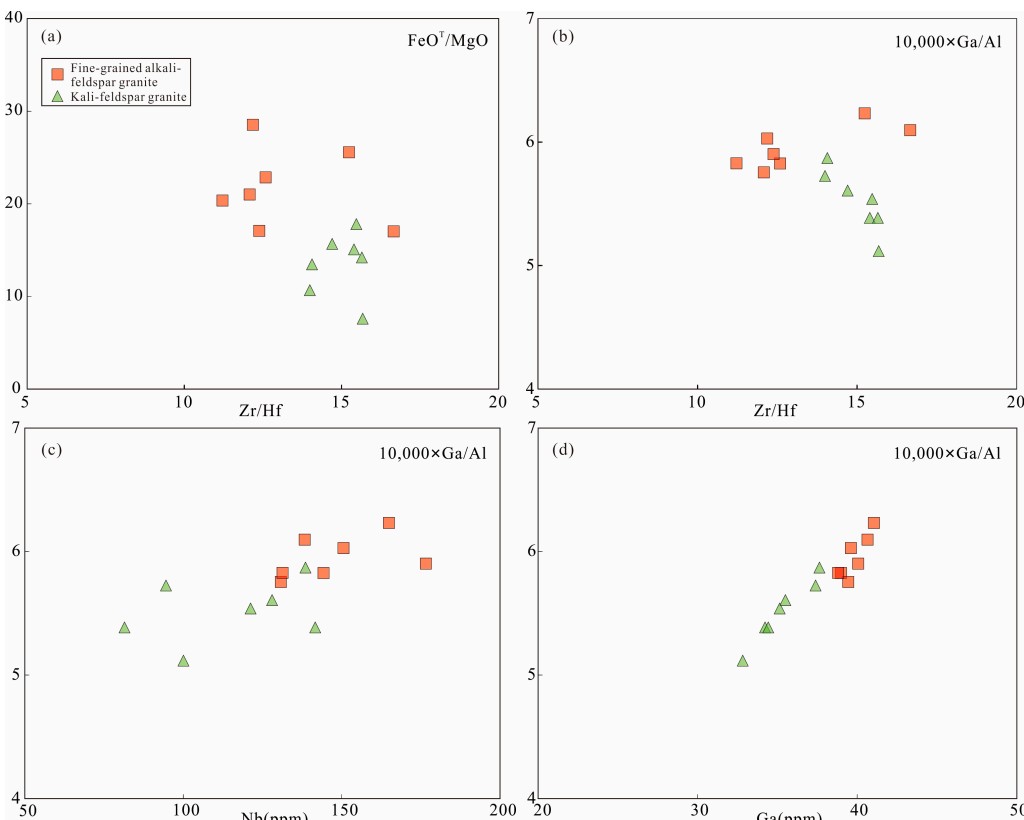

**Figure 11.** Zr/Hf versus FeO$^T$/MgO (**a**), Zr/Hf versus 10,000 × Ga/Al (**b**), Nb versus 10,000 × Ga/Al (**c**), and Ga versus 10,000 × Ga/Al (**d**) discrimination diagrams for evolution degree of granites.

The Dongjin intrusion is obviously rich in silicon and alkali and deficient in Ba, Sr, P, and Ti, showing extremely negative Eu anomalies with a noticeable tetrad effect. These geochemical characteristics all indicate that the magma had experienced high fractional crystallization. With progressive magma differentiation, the obvious loss in Sr content in the residual melts implies the fractional crystallization of plagioclase [72,73]. Low Ba/Sr ratio and Ba depletion suggest that the fractional crystallization of K-feldspar occurred [72]. The fractional crystallization of biotite may be responsible for the enrichment of Nb and Ta contents and the decrease in the Ba/Sr ratio [72,73]. In the Sr versus Rb/Sr and Sr versus Ba diagrams (Figure 12a,b), the contents of Sr and Ba are significantly reduced and the ratio of Rb/Sr is increased from the KG to the FAG, indicating that the fractional crystallization of plagioclase, K-feldspar, and biotite occurred during magmatic evolution.

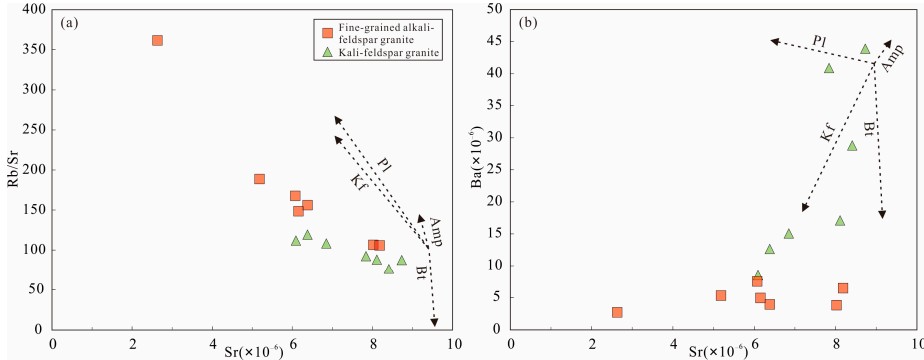

**Figure 12.** Diagrams showing the fractional crystallization of the Dongjin intrusion: (**a**) Sr versus Rb/Sr diagram; (**b**) Sr versus Ba diagram (modified after Qiu [74] and Wang [75]).

The Zr/Hf and Nb/Ta ratios have long been recognized as geochemical "identical twins" that are hardly changed in normal magmatic systems [46,76], but these ratios can change significantly during magmatic differentiation [77,78]. Ballouard et al. [78] suggested that a Nb/Ta ratio of 5 can be divided by the granites into normal crystallization differentiation genesis and magma–hydrothermal interaction genesis, and most granite samples were indicative of undergoing the hydrothermal process when Nb/Ta < 5. Bau [79] proposed Zr/Hf = 26 as the magma–hydrothermal boundary of the granitic system. From the KG to the FAG, the negative Eu anomaly is more pronounced (Figure 7a) and the Zr/Hf ratio decreases (Figure 13a), indicating that the FAG is more evolved than the KG. In the final stages of magmatic evolution, fluid–melt interaction alters the geochemical behavior of rare earth elements in the magma, resulting in a noticeable tetrad effect [31,79–81]. Irber [80] concluded that rocks with the degree of the tetrad effect ($TE_{1,3}$) values exceeding 1.1 have an obvious tetrad effect. $TE_{1,3}$ values of the KG range from 1.18 to 1.24, and $TE_{1,3}$ values of FAG range from 1.23 to 1.29 (Figure 13b,c), indicating that the Dongjin intrusion formed was closely associated with the melt–fluid interaction during the late stage of the high differentiation processes of granitic magma, leading to an obvious tetrad effect. From the KG to the FAG, the tetrad effect is obviously enhanced (Figure 13c,d), and the Y/Ho ratio decreases (Figure 13d).

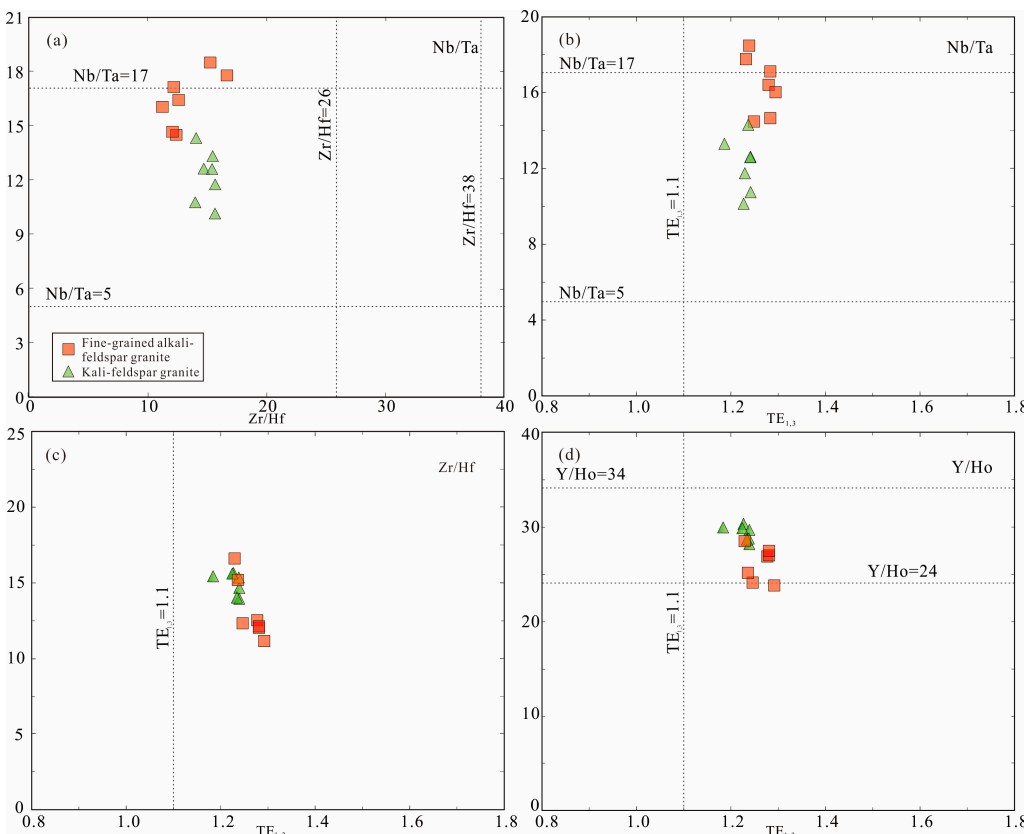

**Figure 13.** Zr/Hf versus Nb/Ta (**a**), $TE_{1,3}$ versus Nb/Ta (**b**), $TE_{1,3}$ versus Zr/Hf (**c**), and $TE_{1,3}$ versus Y/Ho (**d**) discrimination diagrams for evolution degree of granites.

There is a correlation between the magmatic evolution and the mica type: with an increase in magma evolution, the mica will gradually evolve toward zinnwaldite or even lepidolite [72,82–84]. In the mica classification diagram proposed by Tischendorf et al. [72], all the KG samples fall into the protolithionite region (Figure 14a), suggesting that the residual melts are rich in F and Li. In the feldspar classification diagram proposed by Deer et al. [85], the plagioclase in the KG falls into the field of Na-rich albite (Figure 14b). The ore-forming rock mass of granitic Nb-Ta deposits tends to show the characteristics

of peraluminous, and the contents of Nb and Ta increase with magma evolution [86–88]; therefore, the enrichment and mineralization of Nb and Ta elements are closely related to the highly fractionated crystallization of granitic magma. In the early stages of magma evolution, the Dongjin intrusion experiences the fractional crystallization of plagioclase, K-feldspar, and biotite so that the ore-forming materials such as Nb and Ta can preliminarily be enriched. In the late stages of magma evolution, the residual melts contain a large amount of highly volatile components, which is a promoter of the high differentiation evolution of magma. The enrichment of volatile components such as F and Cl can significantly increase the NBO in the melt, thereby improving the solubility of Nb and Ta in silicate melt [46,87]. At the end of magma evolution, ore-forming fluid exsolution occurs in the magma, and the fluid–melt partition coefficients of Nb and Ta are very low [89,90]; therefore, they tend to enrich in the residual melt [91,92]. However, during this period, the fluid–melt interaction is of great significance for the precipitation of Nb and Ta minerals. High-temperature and high-pressure experiments have shown that, at 450 °C, a pegmatitic magmatic system with fluid rich in a fluid-mobile element (FME, e.g., Mn) can crystallize columbite with only 17 ppm Nb and 1 wt% Mn. At the same time, the recent experimental study replicated columbite group mineral textures [92], that is, to a large extent, the interaction of a melt that is enriched in high field strength elements (HFSE) with a Mn-rich fluid can promote the formation of Nb mineralization. Therefore, we believe that fluid–melt interaction is of great significance for the formation of rare metal mineralization in the Dongjin intrusion.

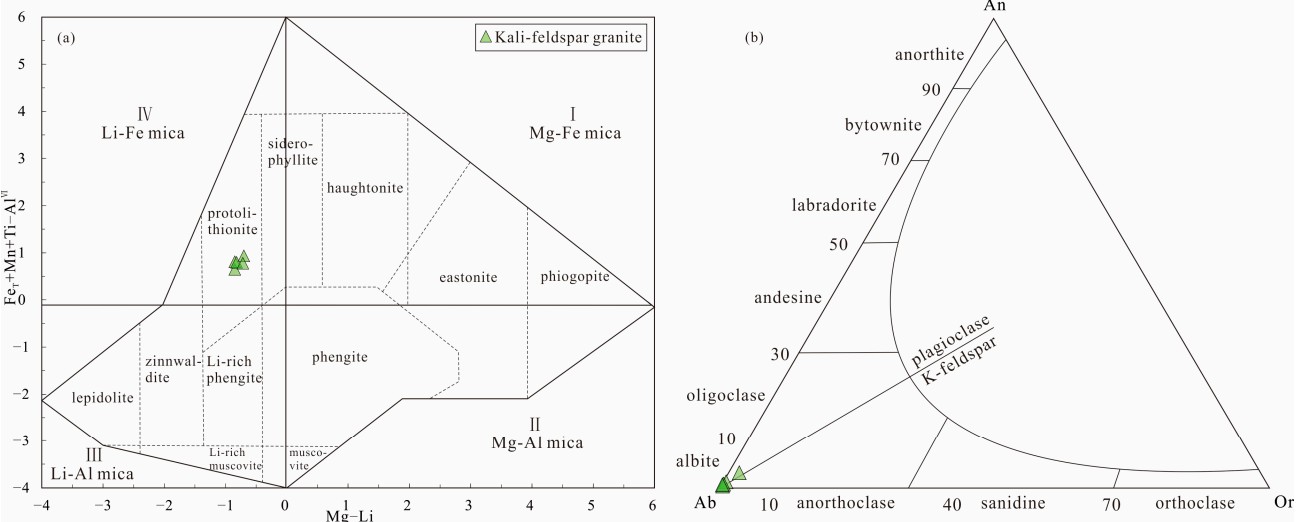

**Figure 14.** (**a**) Diagram of $Fe_T + Mn + Ti - Al^{VI}$ versus $Mg - Li$ for mica minerals (modified after [72]) and (**b**) An-Ab-Or diagram for feldspar minerals.

In summary, the crystallization differentiation of a vast number of rock-forming minerals gradually enriched Nb and Ta in the Dongjin intrusion in the early stages of the magma evolution. The fluid–melt interaction in the final stages of magma evolution led to the large-scale crystallization of Nb and Ta in the residual melts, ultimately forming rare metal mineralization.

### 6.4. Tectonic Setting

At least three stages of tectonic and magmatic activities occurred in the northern margin of the NCC during the Early Carboniferous–Late Triassic, including the Early Carboniferous–Early Permian (358–272 Ma), the Middle Permian (272–259 Ma), and the Late Permian–Late Triassic (259–201 Ma), with striking differences in their tectonic backgrounds.

During the Early Carboniferous–Early Permian period, there were a large number of EW-oriented, high-K calc-alkaline volcanic rocks distributed along the northern margin of the NCC, mainly consisting of hornblende gabbro, diorite, granodiorite, and granite. The

geochemical and source characteristics of these rocks indicate that they were formed in an active continental margin environment [36,93–99]. In the Middle Permian (272–259 Ma), the northern margin of the NCC was in a syn-collision tectonic environment, and the evidence is as follows. The zircon U-Pb age of the Suzy volcanic rocks with characteristics of S-type granite located in the Bayan Obo area is 272–267 Ma, which formed during the transition period from subduction to a syn-collision tectonic environment [100]. Additionally, the closure of the western side of the PAO occurred during this period. Recent paleontological studies indicate that the northern margin of the NCC developed Cathaysia flora along the southern side of the Xar Moron fault belt, while Angara flora developed on the northern side during the Early Permian. The mixture of Cathaysia and Angara flora did not occur until the Middle to Late Permian [101–105]. Paleomagnetic studies have also shown that the Xilinhot–Songliao Block and North China Block reveal a latitudinal convergence and relative rotation between them that led to a scissor-like closure of the PAO from west to east between 265 and 246 Ma [106].

As mentioned above, the Dongjin intrusion and Nb-Ta mineralization were formed in the Early Triassic. During this period, the intrusive rocks in the northern NCC consisted primarily of monzogranite, syenogranite, and monzonite, with minor mafic–ultramafic rocks and granodiorite. The majority of the Early Triassic magmatic rocks are characterized by high contents of $SiO_2$, low initial $^{87}Sr/^{86}Sr$ ratios, and low and negative $\varepsilon Nd(t)$ and $\varepsilon Hf(t)$ values. The granites are categorized as highly fractionated I-type or A-type [107]. Similarly, the Dongjin intrusion belongs to high-K calc-alkaline rocks, characterized by high silicon and high alkalis. In the Yb versus Ta discrimination diagram [108], all samples lie in the within-plate granite field (Figure 15a). In the (Yb + Ta) versus Rb diagram [108], most samples fall into the transition domains of the syn-collisional granite and the within-plate granite (Figure 15b). In the $R_1$ versus $R_2$ discrimination diagram [109], the FAG samples fall into the transition zone between the late-orogenic granite and the post-orogenic granite, and all of the KG samples fall into the post-orogenic granite field (Figure 15c). In the $SiO_2$ versus $\log[CaO/(Na_2O + K_2O)]$ diagram recommended by Brown et al. [110], the FAG samples plot in the area of granite formed near the extensional environment and all the KG samples lie within the extensional granite area (Figure 15d), indicating that the Dongjin intrusion was formed in the transitional field from the syn-collisional granite to the within-plate granite, i.e., the post-collision extensional setting. From the Middle to Late Triassic, alkaline intrusive complexes including nepheline syenite, aegirine–augite syenite, pyroxene syenite, quartz syenite, syenite, alkaline granite, and associate mafic–ultramafic rocks are also quite common in the northern NCC, suggesting an extensional tectonic setting [107]. Furthermore, Late Triassic extension has been reported in the NW Ordos Basin and controlled Triassic sedimentation in the Helan Shan and Zuozi Shan [111,112]. A Late Triassic metamorphic core complex was reported in the south of Sonid Zuoqi in the vicinity of the Solonker suture zone north of the northern margin of the NCC [113]. A Late Triassic NE–SW extension indicated by the L-tectonites exists near the Chifeng area in northeastern NCC, and geo-chronological results on syntectonic diorite plutons and mylonitic rocks indicate deformation at ca. 228–219 Ma [114].

In summary, we believe that the PAO in the northern margin of the NCC was closed during 250–248 Ma. In the Late Permian–Late Triassic, the deformation patterns in the northern NCC changed from a N–S to NE–SW contraction to an extension, and the up-welling of the asthenosphere and the thinning of the crust induced a lot of magmatic activities, including the Dongjin intrusion [36,115–117]. To put it simply, the Dongjin intrusion was formed in a post-collision extensional environment.

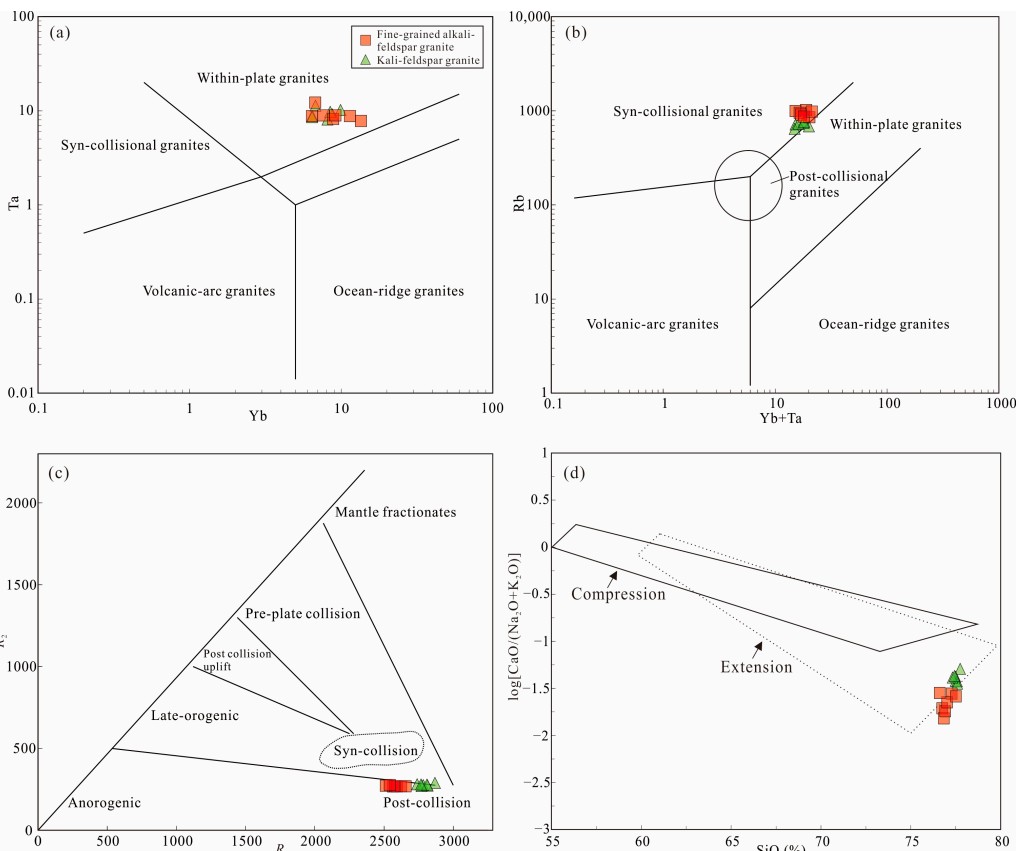

**Figure 15.** Tectonic discrimination diagrams for the Dongjin intrusion: (**a**) Yb versus Ta diagram [108]; (**b**) (Yb + Ta) versus Rb diagram [108]; (**c**) $R_1$ versus $R_2$ diagram [109]; (**d**) $SiO_2$ versus $log[CaO/(Na_2O + K_2O)]$ diagram [110].

## 7. Conclusions

1.  Columbite U-Pb dating for the FAG and KG yielded ages of 248.9 ± 1.9 Ma and 250.1 ± 1.1 Ma, respectively, indicating that the Nb-Ta mineralization occurred in the Early Triassic. There is rare metal mineralization related to highly fractionated granites in the MNNCC in the Early Triassic;

2.  The Dongjin intrusion belongs to a highly differentiated I-type or A-type granite, and the fractional crystallization of plagioclase, K-feldspar, and biotite occurred during magmatic evolution. The Dongjin intrusion also experienced fluid–melt interaction, which showed a noticeable tetrad effect of rare earth elements;

3.  In the early stages of magmatic evolution, the high degree fractional crystallization of the granitic magma leads to an enrichment in Nb and Ta. In the final stages of magmatic evolution, melt–fluid interaction plays a key role in the further enrichment and mineralization of Nb-Ta;

4.  The Dongjin intrusion was formed in a post-collisional extensional environment.

**Author Contributions:** Conceptualization, C.L. and G.C.; methodology, C.L.; software, C.L., K.L., Z.L. and Y.S.; validation, G.C. and J.W.; formal analysis, C.L., Y.C., K.L. and Z.L.; writing—original draft preparation, C.L.; writing—review and editing: G.C. and J.W.; investigation, C.L., G.C., J.W., Y.C., K.L., Z.L. and Y.S.; resources, G.C., J.W. and Y.C.; data collection, G.C. and J.W.; data curation: G.C. and J.W.; visualization, C.L., K.L. and Z.L.; supervision, G.C., J.W. and Y.C.; project administration and funding acquisition, G.C. All authors have read and agreed to the published version of the manuscript.

**Funding:** This research was funded by the Science and Technology Research Project of Universities in Hebei Province (Grant No. BJK2023077), the Youth Fund of the National Natural Science Foundation of China (Grant No. 42202080), the Open Project Program of Hebei Province Collaborative Innovation

Center for Strategic Critical Mineral Research, Hebei GEO University, China (Grant No. HGUXT-2023-6), and the Geological Science and Technology Project of the Hebei Bureau of Geology and Mineral Exploration and Development (Grant No. 13000022P0069B410038J).

**Data Availability Statement:** Data are contained within the article.

**Acknowledgments:** I would like to especially thank my tutors Gongzheng Chen and Jinfang Wang for their guidance, suggestions, and support. At the same time, we would like to thank Han Zhang from the Beijing Yandu Zhongshi Test Technology Co., Ltd., Beijing, China, and Huan Wang from at the Tuoxuan Rock and Mineral Testing Service Co., Ltd. (TRMTS), Langfang, Hebei Province, China, for their help during the testing process.

**Conflicts of Interest:** The authors declare no conflict of interest.

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
