# Peer review of "Age and Petrogenesis of the Dongjin Rare Metal Mineralized Intrusion in the Northern Margin of the North China Craton"

_minerals, doi:10.3390/min13121477_

Round 1

Reviewer 1 Report

Comments and Suggestions for Authors

Review of “Age and petrogenesis of the Dongjin rare metal mineralized intrusion in the northern margin of the North China Craton” submitted by Chenyu Liu et al. to Minerals

Dear Authors and Editors,

Manuscript submitted by Chenyu Liu et al. (2023) to Minerals is a well-structured, detailed petrogenetic study of the rare metal mineralized, highly fractionated Dongjin granitoid intrusion (North China Craton). Besides the formation of the studied granitoid body (applying traditional petrological and geochemical methods), the age (by columbite U-Pb dating) and the formation of the related rare metal mineralization have also been under focus which makes the new results useful regarding economical perspectives, too. Most parts of the study are easy to comprehend, the key questions as well as the regional geology are well-introduced, however, in some sections (e.g., the description of the Results and in some places of the Discussion) the text could be completed or slightly improved and some of the implications reconsidered. My recommendations are listed below (ordered by their significance), while some corrections, regarding the English and the terminology of the manuscript are also listed in the following section of the review. After considering these suggestions and modifying some parts of the text accordingly, I recommend the paper to be published in Minerals.

Comment 1 (lines 174–179 and Fig. 3): The “Petrography” section is very short and missing many important details regarding the modal composition and the texture of the studied granites. Please, give the modal composition of some representative samples (by modal estimation or point counting), e.g., X vol% quartz/K-feldspar/plagioclase/biotite. A short description of each rock-forming mineral is another essential part of the petrography, giving the crystal size (min/max, average) of the minerals, their shape (euhedral/subhedral/anhedral), alterations or inclusions (if there any). The petrographic description should be completed by more information about the accessory minerals. Using the modal composition, petrography-based names should be given to the studied samples (e.g., alkali feldspar granite, syenogranite, according to the Streckeisen diagram). The generally used “fine-grained alkali-feldspar granite” and the “kali-feldspar granites” terms are quite uncommon, I would name the studied lithologies after their modal mineralogy (e.g., alkali feldspar granite, syenogranite), however using their dominant crystals size is informative, as well.

Some recommendations to Fig. 3: (1) add some brightness to the photomicrographs, they are dark in the recent pdf version; (2) use “photomicrographs” instead of “micrographs”; (3) use “crystal size” instead of “particle size”; (4) use “cross polarized light” instead of “orthogonal polarized light”.

Comment 2 (Chapters 6.2 and 6.4): In general, I do not agree to completely rule out the anorogenic (A-type) origin of the studied samples, moreover, it seems to me more feasible than they are fractionated I-type granites. The whole-rock major and trace element geochemistry (e.g., high SiO2 and relatively high alkali contents, low CaO and MgO contents, peraluminous and ferroan characters, “seagull” REE pattern with deep negative Eu anomaly, enrichment in U, Th, Zr, Rb, Nb, Y and depletion in Ba, Sr, Ti, P) is typical for A-type granites as well as the relatively low abundance of mafic minerals (including the lack of amphibole). There is no doubt that the granites are highly fractionated and the rare metal mineralization is associated with it, however, the fractionation might occur to an A-type (anorogenic, but not necessarily alkaline) and not to an I-type silicic melt. The latter is in good correspondence with the obtained geotectonic setting (post-collisional extension) that is most commonly associated with A-type (anorogenic) granitoids. Moreover, direct structural (e.g., mafic enclaves/xenoliths), compositional (e.g., amphiboles, presence of allanite and lack of monazite as accessory minerals) or geochemical (metaluminous character) evidence of the possible I-type origin of these granitoids is missing. To sum up, I would raise two possible, distinct origins for the studied rocks (1) a more feasible A-type (anorogenic) and (2) a highly fractionated I-type origin in the Discussion/Conclusions.   

Comment 3 (lines 259–267): Some relevant information is missing from the results of the columbite dating, e.g., the presence of the discordant dates (if there were any; some may be visible in Fig. 5). If discordant dates were obtained, were they omitted from concordia age calculations? Their discordance values should be also given (e.g., in Table 1 for each spot). The single dates of U-Pb columbite dating (e.g., U238/Pb206, U237/Pb205) are also interesting as well as the age ranges given by them, they should be added to the manuscript, too (e.g., in Table 1 for each spot), because only the calculated concordia ages are available in the recent version.

Comment 4 (lines 292–294 and 304–307): I suggest describing element enrichments and descriptions relative to primitive mantle or chondrite and not to the neighboring elements.

Comment 5 (Chapter 5.3): The text of the EMPA results is very short and the analyses are rather incompletely documented. I suggest adding some photomicrographs (as a new figure) to the manuscript in which some of the analyzed micas and feldspars with the exact sites of the spot analyses are visible.

Comment 6 (lines 81, 92, 328 etc.): I would not use the term “diagenesis/diagenetic” for an igneous body. I recommend replacing it, e.g., by “crystallization ages”.

Comment 7 (Fig. 1): Please, indicate in the figure caption that the presented ages are magmatic (crystallization) ages or the ages of the rare metal mineralizations. In the figure caption, please correct the word “classic” rock.

Comment 8 (Fig. 2): I suggest giving the GPS coordinates of the sampling localities of the study in the figure or in the figure caption.

Comment 9 (lines 193–194): It is not clear from this part of the text that “columbite samples” refers to grain mounts of separated crystals, prepared for U-Pb dating. I would complete the text with this information.

Comment 10 (line 282, 310): Instead of “SiO2 vs. Na2O + K2O diagram” please use total alkali-silica (TAS) diagram.

Comments on the Quality of English Language

Please, find some minor corrections listed below regarding the English of the manuscript and some corrected typos:

Comment 1 (line 72): Instead of “amazonitization and albitization granite” use “amazonitized and albitized granite”.

Comment 2 (line 75): Instead of “highly degree crystallization” use “high degree crystallization”.

Comment 3 (line 144): Instead of “intermediate volcanic lava-volcanic rocks” use “intermediate volcanic rocks” or “intermediate lavas”.

Comment 4 (lines 144, 146, 147, 148 etc.): In the name of the “formations” use capital “F”.

Comment 5 (line 186): Please, correct “quatz” to “quartz”.

Comment 6 (line 200): Instead of “selection” I recommend using “separation”.

Comment 7 (line 205): Instead of “columbite samples” use “columbite crystals”.

Comment 8 (line 251): Please correct “wollatonite” to “wollastonite”.

Comment 9 (lines 291 and 303): Instead of “δEu”, I recommend using “Eu/Eu*” for the Eu anomaly.

Comment 10 (lines 293 and 306): Please correct “T” to “Ti”.

Comment 11 (line 315): Use “spider diagrams” instead of “spider grams”.

Comment 12 (line 348):  Instead of “amazonitization granite” use “amazonitized granite”.

Comment 13 (line 377): Instead of “losses”, I recommend using “depletions”.

Comment 14 (line 391): Use “mafic minerals” instead of “dark minerals”.

Comment 15 (line 548): Use “highly fractionated” instead of “high fractionated”.

Reviewer 2 Report

Comments and Suggestions for Authors

This paper applies columbite U-Pb dating, elements and mineral geochemistry to constrain characteristics of Dongjin granites. The analytical results are authentic and most of the discussions are reasonable. However, there still remain some improper contents in the manuscript (some contradictions or missing parts). In addition, pay attention to English grammars. The charts must be checked carefully. A moderate revision is suggested for its publication in Minerals.

Specific comments:

Replace all figures with high definition pictures. The vertical and horizontal coordinates of the picture and the font in the picture can be larger.

Line 52-53: Please add references to support the conclusion.

Line 85: Add why columbite is chosen to date. In Dongjin granite, whether it has universality, representativeness, deficiency and so on.

2. Regional Geology

This part lacks the introduction of the Nb-Ta mineralization characteristics of Dongjin granite. Please add it.

3. Petrography

The petrological description of this part is too simple, please explain it in detail. You can move the last paragraph of the previous chapter to this section.

5. Results

Please list the full name or meaning or formula of these abbreviations (DI, A/CNK, A/NK, LREE, HREE) in the text or table.

Figure 6a and 6b: Why choose 2 pictures to name Dongjin intrusions? The naming of intrusions should be combined with hand specimens, microscope pictures and geochemical characteristics.

Table 3 and 4: Missing the unit “wt.%”

Figure 10: Missing the unit “ppm”

6.1 rock- and ore-forming ages

1-2 paragraphs are long and meaningless, so you can delete and merge them.

Line 360-362: What does this last sentence mean? Did you solve it after you brought it up? If No, please handle it carefully.

Line370-372: How did you deduce the last sentence? It is not clear that Nb-Ta mineralization must be related to high fractionated granites, and the last sentence of the second paragraph also said that high fractionated granites may also be poor in ore.

6.2 Petrogenetic type

Please carefully consider whether it is high fractionated I-type or A-type granite. You can learn from the literature (Junjie Zhang 2023 Chemical Geology; Saijun Sun 2015 Lithos).

6.3 Magmatic evolution and mineralization

There are contradictions. For example, Line 409-411: Figure 10b shows that the FAG and KG have opposite trend. Refer to previous literature (Junjie Zhang 2023 Chemical Geology; Saijun Sun 2023 Minerals)

Line 349: “The is” shou be “This is”.

“TE1.3”, please give an explanation of the definition in the text or chart.

Why do some highly differentiated granites fail to form Nb-Ta deposits? Who is the key factor between high differentiation and fluid-melt interaction in Nb-Ta deposits?

Line 500: What does “PAO” mean?

Line 536-540: There are contradictions between before and after. What tectonic setting formed the Dongjin granite? This depends on comprehensive factors (age, geochemical feature, sedimentary, structure, etc.), not just on the construction of discriminant graphs.

Comments on the Quality of English Language

Pay attention to English grammars.
